# Reconstruction of a regulated two-cell metabolic model to study biohydrogen production in a diazotrophic cyanobacterium *Anabaena variabilis* ATCC 29413

**Ali Malek Shahkouhi**[◉], **Ehsan Motamedian**[ID][◉]*

Department of Biotechnology, Faculty of Chemical Engineering, Tarbiat Modares University, Tehran, Iran

◉ These authors contributed equally to this work.
* motamedian@modares.ac.ir

## Abstract

*Anabaena variabilis* is a diazotrophic filamentous cyanobacterium that differentiates to heterocysts and produces hydrogen as a byproduct. Study on metabolic interactions of the two differentiated cells provides a better understanding of its metabolism especially for improving hydrogen production. To this end, a genome-scale metabolic model for *Anabaena variabilis* ATCC 29413, iAM957, was reconstructed and evaluated in this research. Then, the model and transcriptomic data of the vegetative and heterocyst cells were applied to construct a regulated two-cell metabolic model. The regulated model improved prediction for biomass in high radiation levels. The regulated model predicts that heterocysts provide an oxygen-free environment and then, this model was used to find strategies for improving hydrogen production in heterocysts. The predictions indicate that the removal of uptake hydrogenase improves hydrogen production which is consistent with previous empirical research. Furthermore, the regulated model proposed activation of some reactions to provide redox cofactors which are required for improving hydrogen production up to 60% by bidirectional hydrogenase.

## Introduction

Cyanobacteria are unique prokaryotes because their oxygenic photosynthesis changed anoxic biosphere to a more oxygen-rich environment about 2.4 billion years ago [1]. Plastids in plants and algae are originated from cyanobacteria through the evolutionary event of endosymbiosis [2] in which they do photosynthesis like higher plants. The metabolic versatility and flexibility of cyanobacteria enabled them to grow in a wide range of habitats such as freshwaters, ponds, wetlands and harsh environments including hot springs, brackish waters, deserts, and cold regions [3, 4]. Cyanobacteria play a key role in providing the primary elements for life including organic carbon, oxygen, and nitrogen [5]. Cyanobacteria need sunlight, water, carbon dioxide, and some minerals to grow, and their photosynthetic efficiency is higher than other oxygenic photosynthetic organisms like algae and plants [6, 7]. Ease of genetic manipulation,

**Data Availability Statement:** All relevant data are within the manuscript and its Supporting Information files.

**Funding:** The authors received no specific funding for this work.

**Competing interests:** The authors have declared that no competing interests exist.

uncomplicated metabolism, and simple cultivation attracted great attention to cyanobacteria as a producer of the third generation of biofuels [8–11] and high-value products [12, 13].

Genotypically and phenotypically related different cell types with dependent growth have complicated life systems [14]. Filamentous cyanobacteria of the order Nostacales have a multicellular metabolically interdependent system [15]. Genus *Anabaena* in this order, are filamentous cyanobacteria that consist of hundreds of photosynthetic vegetative cells, arranged in an unbranched and uniseriate filament [16, 17]. In the absence of nitrate or ammonia in the growth media, vegetative cells undergo heterocyst differentiation enabling nitrogen fixation [16, 18–20]. In the filaments, combined nitrogen depletion stimulates gene expression, and heterocyst cells develop along the filament at the semiregular pattern, nearly 5–10% of the cells [17, 21]. The nitrogenase enzyme which is responsible for the fixation of molecular nitrogen is oxygen labile and the produced oxygen during photosynthesis will damage this enzyme irreversibly [22]. Heterocysts are characterized by having a thick cell wall that limits the entrance of oxygen, deactivated $O_2$-producing photosystem II, and a high respiration rate that scavenges the remaining oxygen [23–25]. Since Ribulose 1,5-bisphosphate (RuBP) carboxylase, a key enzyme in fixing carbon dioxide is absent in the heterocysts, sucrose is transported from vegetative cells to heterocysts as the dominant form of carbon [26, 27]. In return, fixed nitrogen in heterocysts is transported to vegetative cells as amino acids and hence, the entire filament will grow by the intercellular exchange of metabolites [24, 28].

Study on heterocysts is an interesting subject of cell differentiation [25] and simultaneous metabolic modeling of the two cells. Investigation of the effect of intercellular interactions provides a better understanding of differentiation. Genome-scale metabolic models are built based on genomic sequenced data [29]. The constraint-based modeling approaches such as flux balance analysis (FBA) [30, 31] are applicable to predict multicellular phenotypes quantitatively. In recent years, many cyanobacteria metabolic models have been developed and most of them were reconstructed for the unicellular cyanobacteria such as *Synechocystis* sp. PCC 6803 [32–35]. Recently, Malatinszky et al. [36] reconstructed a genome-scale metabolic model for the cyanobacterium *Anabaena* sp. PCC 7120, a heterocyst forming and diazotrophic cyanobacterium. They used FBA to study the effect of intercellular exchange of metabolites between heterocysts and vegetative cells on the optimal growth rate.

Even though genus *Anabena* has garnered interest for its biohydrogen production during nitrogen fixation [37–39], its metabolic models have not been used to study biohydrogen production. *Anabaena variabilis* ATCC 29413 (hereafter *A. variabilis*) has one of the highest hydrogen production rates in cyanobacteria and has been popular candidate for studying biohydrogen production [40, 41]. Therefore, a curated genome-scale metabolic model for *A. variabilis* ATCC 29413 (named iAM957) was reconstructed in this research for the first time to study the metabolism of biohydrogen production. The model and gene expression data for vegetative and heterocyst cells of *A. variabilis* ATCC 29413 [42] were integrated using TRFBA [43] to develop a regulated two-cell metabolic model. One of the major obstacles for sustainable hydrogen production in $N_2$-fixing cyanobacteria is the irreversible inhibition of the three enzymes involved in $H_2$ production (nitrogenase, uptake hup-hydrogenase, and bidirectional Hox-hydrogenase) by oxygen [44] that was studied using the model.

The model was exploited to study metabolic interactions between the two differentiated cells and to find strategies for improvement of hydrogen production. The possibility of providing a condition without oxygen production and consumption in the metabolism of vegetative and heterocyst cells was investigated using the two-cell model to determine which cell is more suitable for $H_2$ production. Then, the model was applied to determine metabolic changes required for redirecting the electron flow to enhance $H_2$ production under suboptimal growth conditions.

## Material and methods

### Reconstruction of a genome-scale metabolic model

The metabolic network of *A. variabilis* ATCC 29413 (iAM957) was reconstructed based on the protocol presented by Thiele and Palsson [45]. The genome annotation and data from Kegg [46], BioCyc [47] and CyanoBase (a specific database for cyanobacteria) [48] were used. A draft network was generated manually and then each reaction was created by using the information available for *A. variabili*s or other cyanobacteria in literature, biochemical databases, and biochemistry textbooks. Mass and charge balances were carried out on each reaction. BiGG [49], Brenda [50], and Uniprot [51] databases were extensively used in the refinement step of the network. Subsequently, transport reactions were added to the network from genome annotation as well as physiological information in the literature. A confidence score was assigned to each reaction in the network based on the available evidence for its presence [45]. Reactions, metabolites and 160 reference papers used for extracting reactions, were presented as an excel file in S1 File.

Calculations were carried out in MATLAB software by the use of COBRA toolbox [52] and the glpk (GNU Linear Programming Kit) package was applied to solve Linear Programming problems.

### Generation of the biomass objective function (BOF)

The growth rate of *A. variabilis* metabolic model was determined by measuring the biomass formation rate and biomass objective function (BOF) was defined for this purpose. The biomass reaction consists of proteins, lipids, DNA, RNA, cell wall components, soluble metabolites, inorganic ions, and pigments. The macromolecular composition of the biomass of *A. variabilis* [53] (in weight percent) is 48.3 protein, 24.53 carbohydrates, 11.6 lipids, 9.1 RNA, 2.27 DNA, and 4.2 ash [53]. Amino acids are the major constituents of the biomass formula, but there is no available experimental data for them. Therefore, the percentage of amino acids was estimated from the genome information according to the method presented in [45]. Glycolipids monogalactosyldiacylglycerol (MGDG), digalactosyldiacylglycerol (DGDG), sulfoquinovosyldiacylglycerol (SQDG), and the phospholipid phosphatidylglycerol (PG) are major lipids of *A. variabilis* [54]. The relative fractions of fatty acids were obtained from experimental work of Naoki et al. [54] for this cyanobacterium. The molar percentage of nucleic acids in DNA and RNA was estimated using the nucleotide sequence of *A. variabilis* [45]. Carbohydrates in biomass composition are glycogen, peptidoglycan, and lipopolysaccharides (LPS). Glycogen is the major carbohydrate in biomass and in carbon starvation conditions, serves as a carbon source for growth. Its fraction in biomass was obtained from Ersnt et al. [55] and the amount of peptidoglycan and lipopolysaccharides was taken from metabolic model of *Synechocystis* sp. PCC 6803 [33]. Chlorophyll a [56], carotenoids [56, 57], and tocopherols [58] are considered as pigments of *A. variabilis* which chlorophyll a has the most portion of pigments in biomass formula. Since, growth-associated ATP maintenance reaction (GAM) and non-GAM reaction (NGAM) for biomass production were not measured experimentally for this organism, the data for *Synechocystis* sp. PCC 6803 model [33] in three auto, hetero, and mixotrophic conditions were used. A detailed procedure for the biomass reaction formula is available in S2 File.

### In silico simulations

Three metabolic models including single-cell, two-cell, and regulated two-cell models were constructed in this research according to the following sections. The mat-files of the single-

cell, two-cell, and regulated two-cell models for implementation in MATLAB are presented in
S3 File.

**Single-cell model.** The single-cell model was used to simulate the growth of vegetative
cells on nitrate as a combined nitrogen source. In the presence of nitrate, heterocyst differenti-
ation does not occur, and hence, the reconstructed metabolic model as a single-cell model was
applied to predict the growth rate using FBA. This method expresses a metabolic model as a
linear programming problem, as in Eq (1), and defines an objective function (Z).

$$\text{Max } Z = cv$$
$$\text{Such that } S.v = b \tag{1}$$
$$v_{min} \leq v \leq v_{max}$$

where c is the vector of the objective function coefficients and S is an m×n matrix with
rank $\leq$ m. m = 921 is the number of equations derived from the metabolic model using the
mass balance for each metabolite and n = 983 is the number of unknown reaction fluxes. b is
the right-hand side vector determined by known reaction fluxes, and $v_{min}$ and $v_{max}$ are the
lower and upper bounds of the variable fluxes, respectively. Upper and lower bounds of all
intracellular reversible reactions were set to 1000 and -1000 mmol/gDCW/h, respectively.
Fluxes were limited between 0 and 1000 mmol/gDCW/h for all intracellular irreversible reac-
tions. Upper and lower bounds of all exchange reactions were set to 1000 mmol/gDCW/h and
zero respectively. Lower bound of exchange reactions for $H^+$, $H_2O$, K, $SO_4$, Pi, and inorganic
ions was set to -1000 mmol/gDCW/h and lower bound of exchange reactions for $HCO_3$, $NO_3$,
photon was set to -10 mmol/gDCW/h. Maximum uptake rate of each carbon source was deter-
mined according to the method presented in [36].

The biomass reaction was used as the objective function to be maximized using FBA and
the single-cell model was compared with the metabolic model of *Anabaena* sp. PCC 7120 [36]
for prediction of growth on different carbon sources. So, according to the method presented in
[36], mixotrophic growth rate for each carbon source was calculated and divided by autotro-
phic growth rate on bicarbonate. Correlation between the predicted relative growth rates
and experimental values for the two models was determined using the Pearson correlation
coefficient.

**Two-cell model.** The two-cell model was constructed to compare its growth predictions
under autotrophic and diazotrophic conditions with those predicted by the regulated two-cell
model and predicted by the previous two-cell model [36]. Upper and lower bounds of all intra-
cellular reactions for the heterocyst and vegetative metabolic models were the same as those
for the single-cell model, except those presented in Table A in S4 File. As nitrogenase is inhib-
ited irreversibly by oxygen and concentration of oxygen in vegetative cells is high [22], it was
removed from the metabolic model of the vegetative cell. Reactions RBCh, RBPC, HCO3E,
GLMS, and PSII are only present in vegetative cells according to the references presented in
Table A in S4 File). Oxygen evolving photosystem II (PSII) is inactivated in heterocyst [25]
because nitrogenase is oxygen-sensitive and in the presence of $O_2$ will be deactivated irrevers-
ibly. Besides, the absence of the key enzyme of the reductive pentose phosphate pathway,
ribulose 1,5-bisphosphate carboxylase (RBCh and RBPC), in heterocyst has been reported in
the literature before [26, 27]. Presence of glutamine-2-oxoglutarate aminotransferase (Fd-
GOGAT) which is the producer of glutamate from 2-oxoglutarate and glutamine in hetero-
cysts of *A. variabilis* was doubtful before [59], but later it was clearly concluded that heterocysts
lack this enzyme [60].

Since ribulose 1,5-bisphosphate carboxylase is absent in heterocysts of *A. variabilis*, fixed
carbon in the form of sucrose transfers from nearby vegetative to heterocyst cells as carbon

source. In return, heterocysts provide the nitrogen source needed for the metabolism of vegetative cells. Glutamate transports from vegetative cells to heterocysts and the fixed nitrogen in the form of glutamine with an additional nitrogen atom than glutamate with the same carbon skeleton shuttles back to vegetative cells as the nitrogen source for growth [61]. Hence, the entire filament can grow by incorporating fixed nitrogen provided in heterocysts. The exchange of these three metabolites (glutamine, glutamate, and sucrose) between the two cell types happens through a continuous periplasm [19, 62].

Upper and lower bounds of all exchange reactions for the two-cell model were the same as those for the single-cell model, except for $HCO_3$, $N_2$, $NO_3$, and photon. Uptake of $HCO_3$ was only considered for vegetative cells and consumption of $N_2$ was only considered for heterocysts. To simulate the diazotrophic condition, uptake of $NO_3$ was limited to zero. Considering both vegetative and heterocyst cells can uptake photon, Eq 2 was added to the model. It should be mentioned that a slack variable was added to the right-hand side of Eq 2 to transform the inequality constraint into equality.

$$v_{p, V} + v_{p, H} \leq v_{p, Max} \tag{2}$$

$v_{p,max}$ indicates the maximum uptake rate of photon, and $v_{p,V}$ and $v_{p,H}$ are photon uptake rates of vegetative cells and heterocysts, respectively. The maximum uptake rates of $HCO_3$ and photon were set to experimental values reported by [63].

The biomass reaction of the vegetative cells was used as the objective function to be maximized using FBA. The biomass of heterocyst cells is 5–10 percent of vegetative cells biomass [16, 22]. In this research, it was assumed that biomass composition of heterocysts is the same as vegetative cells and heterocysts comprise 10 percent of the entire filament and therefore, Eq 3 was added to the two-cell model.

$$v_{g, H} = 0.1 \times v_{g, V} \tag{3}$$

$v_{g,V}$ and $v_{g,H}$ are growth rates of vegetative and heterocyst cells, respectively. The calculated biomass objective function was multiplied by 1.1 for two-cell models to report the total growth rate. The two-cell model includes 1970 variables including reaction fluxes of the two cells, three transport reactions (for intercellular exchange of sucrose, glutamine, and glutamate), and a slack variable for Eq 2. This model also includes 1844 equations including 1842 equations from the two metabolic models and Eqs 2 and 3. Prefixes Vegetative and Heterocyst were used to indicate metabolites and reactions of the vegetative and heterocyst models, respectively. It should be mentioned that Eqs 2 and 3 were also added to the two-cell model presented by Malatinszky et al. [36].

**Regulated two-cell model.** The TRFBA algorithm presented by Motamedian et al. [43] was applied to integrate gene expression data [42] of vegetative and heterocyst cells in photo-autotrophic and heterotrophic conditions for *A. variabilis* ATCC 29413 with the two metabolic models. This algorithm first converted the metabolic model to irreversible and "without OR" form that increased the number of variables to 3669 for the two-cell model. Then, TRFBA added a constraint for each metabolic gene with measured expression level to the two-cell model using Eq (4).

$$\sum_{i \in K_j} v_i \leq E_j \times C \tag{4}$$

where $K_j$ denotes the set of indices of reactions supported by metabolic gene j, $v_i$ is the flux of reaction i, and $E_j$ is the expression level of $j^{th}$ gene. C is a constant parameter that converts the expression levels to the upper bounds of the reactions. The unit for C is mmol gDCW$^{-1}$h$^{-1}$ and

this coefficient indicates the maximum rate supported by one unit of the expression level of a gene. The optimal value of C was determined using sensitivity analysis. By adding a positive slack variable ($a_j$) for each gene to Eq 4, the inequality constraint is transformed into an equality constraint (Eq 5).

$$\sum_{i \in K_j} v_i + a_j = E_j \times C \tag{5}$$

The expression level of 951 metabolic genes was measured for heterocysts and vegetative cells and hence, 1902 variables and equations were added to the regulated two-cell model. The added equations and variables were named with their corresponding gene and prefix Vegetative or Heterocyst. The maximum uptake rate of $HCO_3$ was set to 1 mmol/gDCW/h and the maximum uptake rate of $N_2$ and photon was set to 1000 mmol/gDCW/h to simulate photoautotroph and diazotrophic growth condition. The maximum uptake rate of $HCO_3$ and photon was set to experimental values [63] for comparison with the two-cell models. To simulate the heterotrophic growth condition using the regulated two-cell model, fructose as carbon and energy source with the maximum uptake rate of 1 mmol/gDCW/h was selected as carbon and energy sources for vegetative cells instead of $HCO_3$ and photon.

Shadow price was also calculated for each metabolic gene to determine the gene that controls the growth rate of *A. variabilis* in high irradiance. Shadow price shows sensitivity of the objective function to change in the right-hand side vector of Eq (5) according to Eq 6.

$$\text{Shadow price} = \frac{\partial v_{g, V}}{\partial(E_j \times C)} = \frac{\partial v_{g,V}}{C \times \partial(E_j)} \tag{6}$$

Shadow price indicates the effect of one unit increase in gene expression on the predicted growth rate of *A. variabilis*. Its positive value for a gene shows that the expression level of the gene is not adequate and this level as an intracellular constraint controls the metabolism for growth.

Maximization of biomass and hydrogen production was used as the objective function for calculations using the regulated two-cell model. Maximal and minimal hydrogen secretion rates were calculated for suboptimal conditions for a fixed growth rate of 90% which is optimal growth and their flux distributions were compared to find strategies for improvement of hydrogen production. To avoid the well-known degeneracy of solutions, the Manhattan norm of the flux distributions was minimized while the growth and hydrogen production rates were bound to the intended values.

## Results and discussion

### The metabolic network of *A. variabilis* ATCC 29413

A draft metabolic network of *A. variabilis* based on genome annotation and data presented in relevant databases was reconstructed manually and then was refined. The reconstruction process has been discussed in the material and methods section. The final model includes central metabolic pathways such as glycolysis, pentose phosphate pathway (PPP), incomplete tricarboxylic acid (TCA) cycle, and Calvin-Benson cycle. In addition, it consists of pathways for biosynthesis of lipids, amino acids, cofactors, vitamins, nucleotides, pigments, and reactions involved in hydrogen metabolism. The resulting single-cell metabolic model (iAM957) includes 960 reactions and 912 metabolites as well as 957 genes that encode enzymes which are responsible for these reactions. The reactions take place in four compartments including

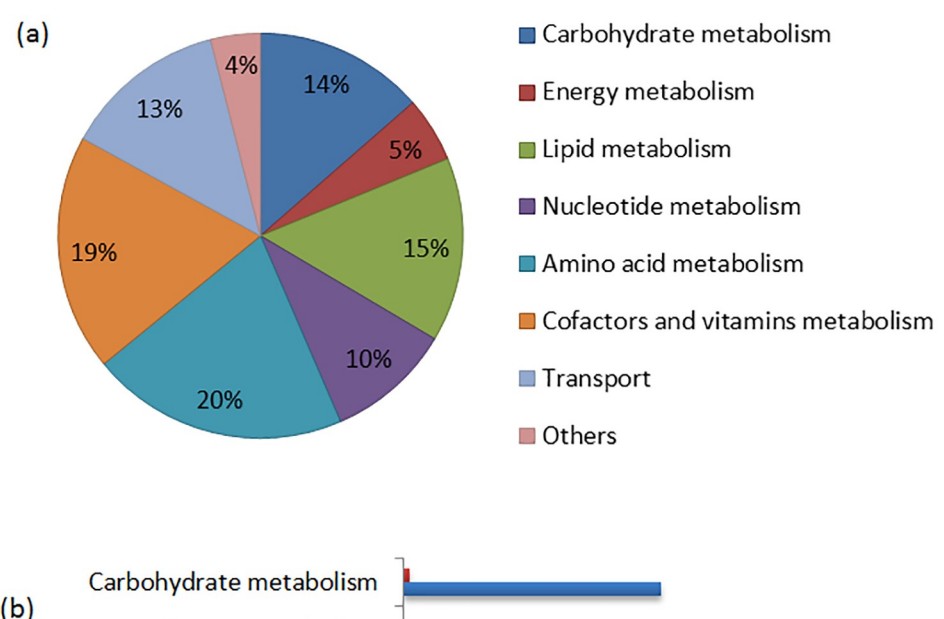

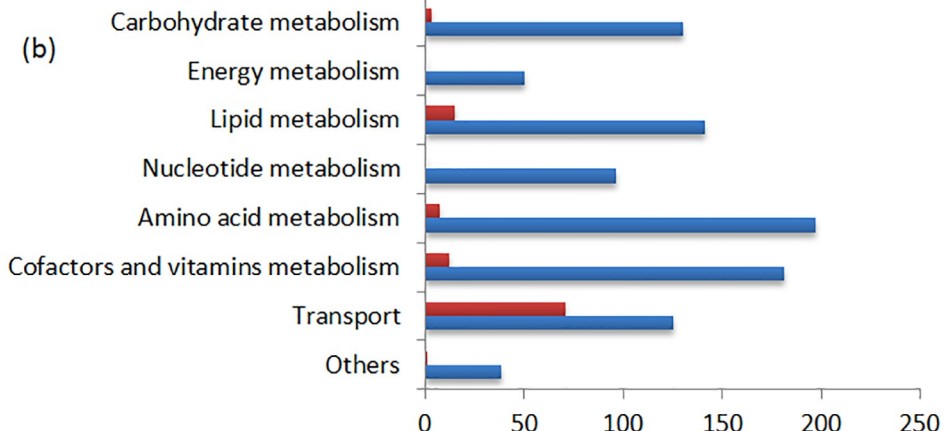

**Fig 1. Properties of iAM957.** (a) distribution of the metabolic reactions in the categories. (b) number of non-gene associated (red) and gene associated (blue) reactions in each category.

cytoplasm, thylakoid, periplasm, and extracellular. Reactions in the network are divided into eight categories comprising 47 pathways, the distribution of which is shown in Fig 1(a). 113 reactions out of 960 are non-gene assigned reactions but they are present in the model. This happens for two reasons; either there was biochemical data confirming their occurrence in the metabolism, or they were necessary for growth. Fig 1(b) shows both the number of genes assigned and non-gene assigned reactions in each category.

A summary of iAM957 and a comparison between the metabolic models of *A. variabilis* ATCC 29413, *Synechocystis* sp. PCC6803, PCC 7942, *Spirulina platensis*, and *Anabaena* sp. PCC 7120 is presented in Table 1. It can be seen that the features of the reconstructed metabolic models for cyanobacteria are different. For example, a comparison between iAM957 and iJN678 demonstrates that iAM957 has more reactions. It is rational considering that *A. variabilis* has a larger genome. There are 276 reactions in iAM957 that are not present in iJN678 and there are 158 reactions in iJN678 that are not present in iAM957. Fig A in S4 File indicates that the different reactions are distributed in various pathways.

**Table 1. Features of the metabolic model reconstructed for *A. variabilis* ATCC29413 in comparison with metabolic models of other cyanobacteria.**

| Features | iAM957 (This study) | iJN678 [33] | iSyf715 [85] | Anabaena [36] | iAK692 [86] |
|---|---|---|---|---|---|
| Species | *A. variabilis* ATCC29413 | *Synechocystis* sp. PCC 6803 | *Synechococcus* elongatus PCC7942 | *Anabaena* sp. PCC 7120 | *S. platensis* |
| Reactions | 983 | 863 | 895 | 897 | 875 |
| metabolic reactions | 825 | 706 | 851 | 818 | 699 |
| other reactions | 161 | 157 | 44 | 79 | 176 |
| metabolites | 926 | 795 | 838 | 777 | 837 |
| Included genes | 957 | 678 | 715 | 862 | 692 |
| Total genes | 5772 | 3725 | 2906 | 5368 | 6176 |
| Percent of coverage | 16.6 | 18.2 | 24.6 | 11.1 | 11.2 |
| Subsystems | 47 | 54 | - | 38 | - |

## Evaluation of the single-cell and two-cell models

**Growth prediction and effect of heterocyst frequency.** The correlation coefficients between the predicted and measured relative growth rates for two single-cell models are presented in Table 2. The predicted rates by both models significantly correlate with the experimental values but the correlation for the single-cell model of this research is a little higher than that for Malatinszky et al. [36]. The predicted relative growth rates by two single-cell models were also compared with experimental values in Table B in S4 File. It can be seen that both models present appropriate predictions of the mixotrophic growth on various carbon sources relative to the autotrophic growth on bicarbonate.

Both two-cell models predict that the increase of the heterocyst percentage slightly decreases the growth rate Fig B in S4 File. Previous researches confirm the predicted effect of heterocyst frequency. Sallah et al. [64] demonstrated that adding an amino acid analog (DL-7-azatryptophan) into the growth medium of *A. variabilis* ATCC 29413 increased the heterocyst frequency and conversely, reduced the growth. Furthermore, Berberoğlu et al. [65] experimentally indicated that the growth rate of *A. variabilis* ATCC 29413 in absence of nitrate is smaller than when nitrate is present.

**Prediction of autotrophic growth.** The comparison of predicted autotrophic growth rates for the two-cell models using FBA and regulated two-cell model using TRFBA with experimental data [63] is illustrated in Fig 2. Sensitivity analysis of the relative error in the prediction of growth rate with respect to the C values was applied to determine the optimal value of 0.05 mmol/gDCW/h for the C parameter of the TRFBA algorithm Fig C in S4 File.

It illustrates that TRFBA and FBA predictions in low irradiance are approximately the same and close to the experimental data while FBA for both two-cell models overestimates the growth rate under high irradiances and bicarbonate uptake rates. FBA does not generally apply any intracellular constraint and only extracellular constraints including bicarbonate and photon uptake rates control the predicted growth rates. Overprediction for high irradiances and uptake rates indicates that intracellular constraints are inhibiting the growth and so,

**Table 2. The Pearson correlation between the predicted and measured relative growth rates on various carbon sources for the two single-cell models.**

| | Single-cell model | |
|---|---|---|
| | **This research** | **Malatinszky et al. [36]** |
| Pearson correlation coefficient | 0.87 | 0.79 |
| P-value | $1.24 \times 10^{-4}$ | $1.3 \times 10^{-3}$ |

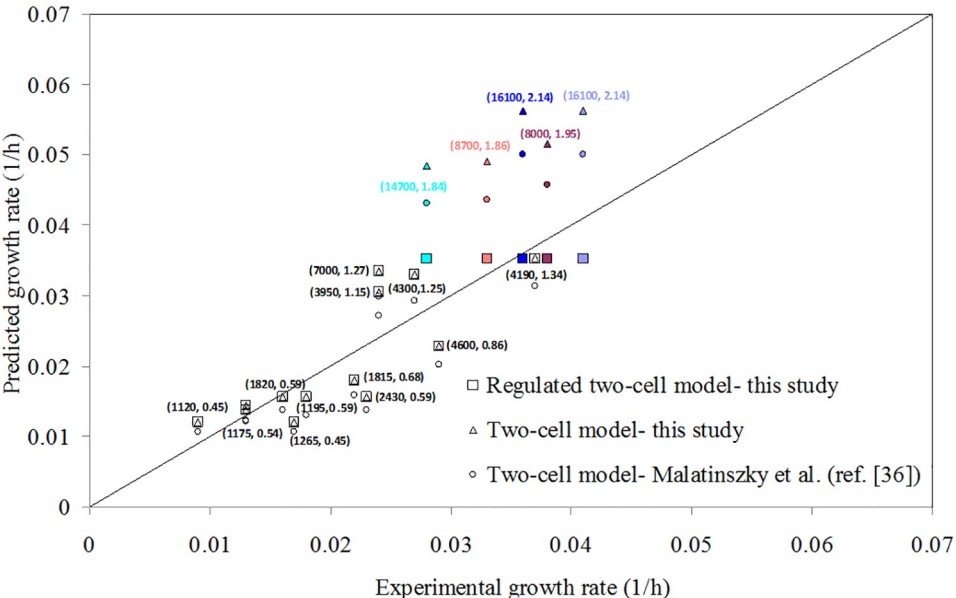

**Fig 2. Comparison of growth rates predicted by the two-cell models and the regulated two-cell model with experimental data** [63]**.** The labeled values indicate irradiance (lux) and bicarbnate uptake rate (mmol/gDCW/h), respectively, for each data. The same colors were used for distant points with the same experimental data.

expression data were integrated with the two-cell metabolic model using TRFBA to impose the intracellular constraints and to find the origin of the inhibition. TRFBA limits the upper bound of intracellular reactions based on the expression level of their corresponding genes. The calculated shadow prices for metabolic genes indicate that the growth rate is sensitive to the expression level of Ava_1852 in the vegetative cells. This gene is responsible for synthesis of protein PsbJ that is one of the components of the core complex of photosystem II. The mean expression level of Ava_1852 is 56 that is significantly less than the average expression level for vegetative cells. Hence, TRFBA predicts that PSII is limiting reaction which controls growth in high levels of radiation and bicarbonate uptake. Shadow price for other genes was zero and hence, the regulated two-cell model predicted that the measured expression level of other genes was adequate for growth. Fig D in S4 File presents the predicted growth rate versus expression level for some genes of photosystem II using the regulated two-cell model. It can be seen that only the expression level of PsbJ was not sufficient for optimal growth. The inhibited growth of *A. variabilis* at high irradiance has also been stated by Yoon et al. [66, 67]. It is also expressed that high light intensities destroy photosystem II reaction centers and lead to photo-inhibition on the growth of the cyanobacterium [68, 69].

Furthermore, the predicted changes in the intercellular exchange fluxes under various auto-trophic growth was evaluated using experimental data of Berberoğlu et al. [63]. No constraint was imposed on the intercellular exchange fluxes of the three two-cell models. FVA [70] was used to calculate the flux variability of each intercellular exchange reaction. The two-cell model of Malatinszky et al. [36] predicted that both sucrose and glutamate can be used as the main carbon source (Fig E(a) in S4 File). However, our two-cell model predicts that sucrose is transported to heterocysts as carbon source in all conditions (Fig E(b) in S4 File). Experimental reports also suggest that the main carbon source for heterocysts is sucrose [71].

Interestingly, glutamate and sucrose intercellular exchange reactions are coupled in the two-cell model of Malatinzsky et al. [36] and the sucrose rate is maximized when glutamate is

exchanged at the maximum rate. Furthermore, Fig E in S4 File shows that maximum exchange rates for this two-cell model are always higher than those for our two-cell models and the rates unreasonably increase with an enhancement of irradiance. The maximal sucrose uptake rate of 46.8 mmol/gDCW/h is calculated at an irradiance of 16100 lux and bicarbonate uptake rate of 2.14 mmol/gDCW/h while this rate for our two-cell is 1.29 mmol/gDCW/h. This indicates that there is an internal cycle of carbon in the two-cell model of Malatinzsky et al. [36]. This cycle circulates carbon sources between heterocyst and vegetative cells and the required energy for this circulation is provided by uptake of photon. Comparison between flux distributions for maximum and minimum intercellular exchange rates indicates that many reactions are involved in generating this internal cycle. Hence, Malatinzsky et al. [36] fixed the glutamine-glutamate exchange ratio to one that results in the reduction of the effect of this cycle.

The regulated two-cell model also transports sucrose to heterocyst in low irradiance (Fig E (c) Fig in S4 File). But both sucrose and glutamate can be used as the main carbon source at high irradiance. This result is rational because the model predicts that the growth rate is constant at high irradiance because of the limitation of PSII (as previously explained) and given the extra bicarbonate available, none of the sources are preferred for carbon transport.

### Analysis of *A. variabilis* for hydrogen production

**Selection of the appropriate cell for hydrogen production.** The major obstacle for producing hydrogen using photosystems is the sensitivity of nitrogenase and hydrogenase enzymes to oxygen [72]. For this purpose, the two-cell model was applied to analyze the simultaneous effect of oxygen and hydrogen exchange on photoautotrophic growth using vegetative and heterocyst cells of *A. variabilis*. Fig 3(a) shows that more hydrogen production in vegetative cells increases oxygen production at a constant growth rate. In fact, when the cell is growing in photoautotrophic conditions, the model predicts that providing an oxygen-free environment in vegetative cells is not possible. Unlike the vegetative cells, Fig 3(b) reveals that more hydrogen production reduces oxygen consumption in heterocysts at a constant growth rate. That is; hydrogen production reduces the growth rate at a constant oxygen exchange rate for both cell types.

The model's prediction is in agreement with the literature [72] in which heterocyst provides an oxygen-free environment for hydrogen production. Furthermore, experimental data of Berberoğlu et al. [65] showed that larger hydrogen production rates are obtained for a medium including higher heterocyst frequency. The spatial separation achieved through the differentiation of heterocyst overcomes oxygen sensitivity issues. As a result, heterocyst is a more suitable host for hydrogen production in the photoautotrophic conditions.

The reason for different metabolisms of vegetative cells and heterocysts is that oxygen-evolving photosystem II in heterocyst is inactivated [73, 74] and the model also predicts that oxygen flux from the environment into heterocyst is very low (see Fig 4). The predicted consumption and production rate of the cytoplasmic oxygen molecule in the metabolism of heterocyst is $8.45 \times 10^{-4}$ mmol/gDCW/h which is much less when compared to the vegetative cell (1.07 mmol/gDCW/h). Besides the fact that glutamate has one more oxygen atom than glutamine, glutamine-glutamate exchange and sucrose transfer to heterocyst provide oxygen atom without a need for $O_2$ uptake. Jensen and Cox [74] measured oxygen concentration in the heterocysts of *A. variabilis* and concluded that a little amount of $O_2$ is necessary to be entered into the heterocyst for providing the energy through respiration which is in agreement with the metabolic model. The diffusion of oxygen to heterocyst through its cell envelope is very low because of the glycolipid and polysaccharide layers and the remaining oxygen is maintained close to zero by respiration [73]. Low $O_2$ demand in the metabolism in addition to low $O_2$

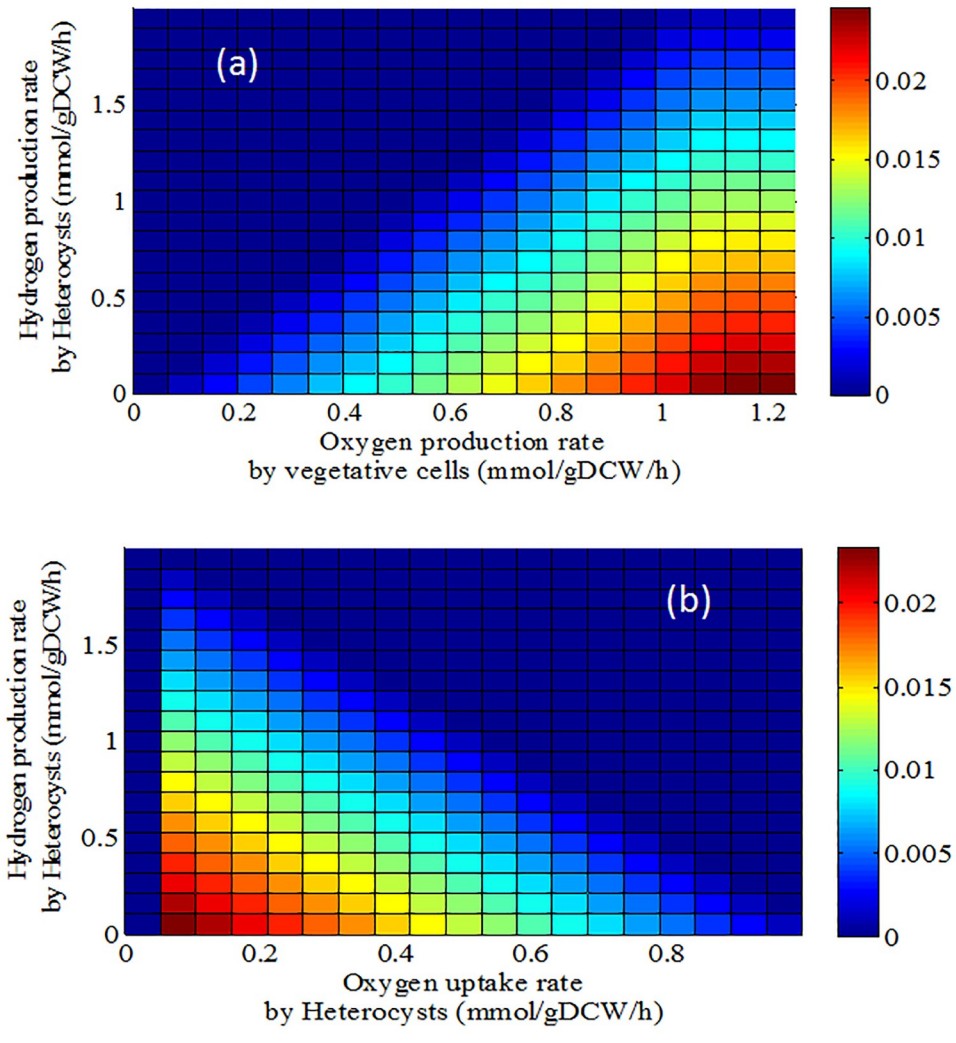

**Fig 3. Double robustness analysis using the regulated two-cell model for photoautotrophic growth of *Anabaena variabilis*.** The heatmap presents changes in the growth rate with varying oxygen exchange and hydrogen production fluxes in a) vegetative and b) heterocyst cells.

diffusion provides an anaerobic environment for the oxygen-sensitive nitrogenase and hydrogenases to be active in the heterocyst.

The model predicts that interaction between hydrogen and oxygen in vegetative cells for heterotrophic growth on fructose is similar to what was predicted for heterocyst in photoautotrophic conditions (Fig F in S4 File). This result is rational for every metabolic model of cyanobacteria because fructose provides both energy and carbon sources for the cell in heterotrophic conditions. However, in photoautotrophic conditions, bicarbonate is used as a carbon source and photon uptake that is necessary for energy generation produces oxygen. Some researchers proposed a two-stage operation for hydrogen production which is in agreement with the model [72]. At the first stage, cyanobacteria grow photoautotrophically and glycogen is accumulated. At the second stage, hydrogen is produced anaerobically by switching from photoautotrophic metabolism to dark fermentation and consuming accumulated glycogen.

**Improvement of hydrogen production.** The intact cells of nitrogen-fixing cyanobacteria produce net hydrogen in small amounts [75]. Thus, the two-cell metabolic model was applied

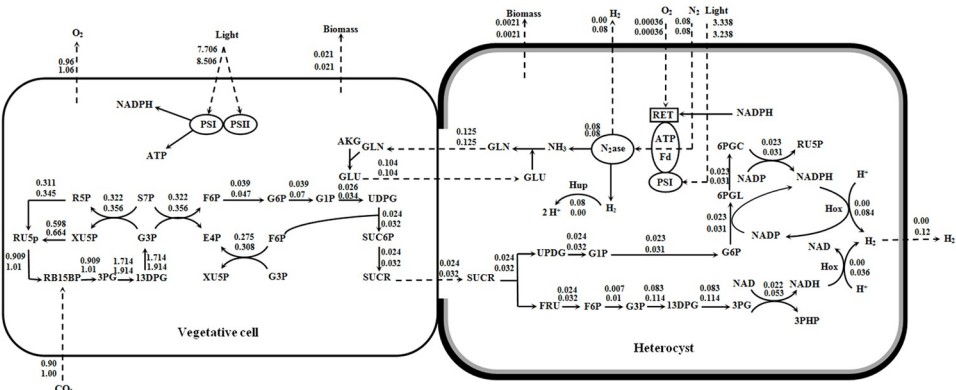

**Fig 4. The predicted flux distributions for minimal (top) and maximal (down) hydrogen production under a constant growth rate.** Abbreviations of metabolites are available in S1 File.

to find improvement strategies for hydrogen production by heterocysts in photoautotrophic and diazotrophic conditions. The model predicts that hydrogen is not produced at the optimal growth rate of *A. variabilis* because of the fact that the increase in hydrogen secretion amount is not in accordance with further growth rate (Fig 3(b)). Thus, the objective function of the regulated two-cell model was changed to hydrogen production in the suboptimal condition of the 90% optimal growth rate to identify pathways and reactions which are effective on hydrogen production in heterocysts. To determine reactions involved in improving the hydrogen production process, flux distribution for maximal and minimal hydrogen productions was compared to each other. The list of up- and down-regulated reactions and their fluxes during maximal and minimal hydrogen production is available in S5 File.

The main reactions that have been up- and down-regulated for improvement of hydrogen production proposed by the two-cell model are shown in Fig 4. In the vegetative cell, it can be seen that the activation of ribulose-bisphosphate carboxylase is increased and more 3-phosphoglycerate has been produced. Successive reactions of the Calvin-Benson cycle and glycolysis pathway have resulted in the overproduction of fructose 6-phosphate and UDP-glucose that in turn are converted to sucrose during the pathway of sucrose metabolism. Required energy and redox cofactors for increasing the sucrose production are provided via the activation of photosystems I and II.

Sucrose, as the carbon source, is transported from vegetative cell to heterocyst and is converted to fructose and UDP-glucose via a reverse reaction catalyzed by alkaline invertase. The experimental results obtained by Schilling and Ehrnsperger [76] confirms the presence of sucrose synthase in vegetative cells and of alkaline invertase in heterocysts. In heterocyst, in a reverse process compared to the vegetative cell metabolism, UDP-glucose is converted to glucose 6-phosphate using UDP–glucose pyrophosphorylase and phosphoglucomutase, then fructose is converted to fructose 6-phosphate by fructokinase.

The two-cell model utilizes glucose 6-phosphate dehydrogenase (G6PDH2), 6-phosphogluconolactonase, and phosphogluconate dehydrogenase from the oxidative pentose phosphate pathway (OPPP) to change glucose 6-phosphate to ribulose 5-phosphate. In the model, enzymes of OPPP and also hexokinase are active and have fluxes, which is in agreement with the results published by Winkenbach and Wolk [77] for *Anabaena cylindrica*. Summers et al. [78] demonstrated that reactions of OPPP and G6PDH are the essential catabolic routes for providing redox cofactors for nitrogen fixation and respiration in differentiated heterocysts and this is what the model confirms properly.

Utilization of OPPP produces two moles of NADPH per mole of glucose 6-phosphate, and then ribulose 5-phosphate is converted to glyceraldehyde 3-phosphate using the non-oxidative pathway of PPP. NADPH is used to produce hydrogen by applying NADPH-linked bidirectional Hox hydrogenases in maximal condition, for producing ATP in respiratory electron transport (RET), and also transferring electrons from ferredoxin in PSI. On the other hand, nitrogenase employes ATP and ferredoxin to produce ammonium and hydrogen. Ammonium activates the glutamine-glutamate shuttle to provide the nitrogen source to the vegetative cell. Fig 4 shows that this shuttle is not applied to improve hydrogen production, and the rate of nitrogenase activity does not change in the maximal and minimal conditions. In fact, Hup hydrogenase consumes the produced hydrogen by nitrogenase to generate reduced ferredoxin. Hup hydrogenase reutilizes the evolved hydrogen by nitrogenase so that almost hydrogen production would not be detectable [79, 80]. Furthermore, removal of Hup hydrogenase improves hydrogen production significantly [81, 82].

The model proposes another less-known pathway for the production of hydrogen. Fructose 6-phosphate produced by fructokinase is converted to 3-phosphoglycerate through glycolysis pathway. Afterwards, 3-phosphoglycerate is converted to formate and $CO_2$. This process provides NADH that in turn is used to produce hydrogen using NADH-linked Hox hydrogenases in maximal condition.

In fact, the two-cell model proposes that the activation of NAD(P)H-linked Hox hydrogenases and the other optimal pathways supply required NADH and NADPH for the production of 60% of hydrogen using Hox hydrogenase in the maximal condition. The high theoretical potential of Hox hydrogenase to produce hydrogen has been investigated [79]. Hox hydrogenase has the capacity of both re-oxidizing and producing hydrogen. However, its removal results in less hydrogen production [82]. Moreover, it is the Hup hydrogenase, and not the bidirectional enzymes which is the most effective element in recycling hydrogen [80]. These results indicate that Hox hydrogenase act as a hydrogen producer in *A. variabilis*. It is worthy to mention that the metabolic energy demand for Hox hydrogenase is less than nitrogenase activity to produce hydrogen [75]. Nitrogen fixation is an expensive reaction energetically that consumes 16 ATP molecules for the reduction of $N_2$ and generates only one molecule of $H_2$ [72, 83]. Besides of the fact of the high energy demand for nitrogen fixation, the nitrogenase turnover rate is 6.4 s$^{-1}$ which is very slow in comparison with the turnover rate of Hox hydrogenase (98 s$^{-1}$) [72, 84]. Considering this issue that Hox hydrogenase in cyanobacteria primarily functions as a redox regulator for maintaining a proper oxidation/reduction state in the cell [79], we should perceive that activation of the proposed optimal pathways provide the required redox cofactors for significant improvement of hydrogen production by Hox hydrogenase.

## Conclusion

In this study, a curated genome-scale metabolic network for heterocystous cyanobacterium *A. variabilis* ATCC 29413 was reconstructed using the information presented in the databases and research papers. iAM957 was used to construct the single-cell for simulation of growth in the presence of nitrate and the two-cell model for simulation of diazotrophic condition. The models were compared with the single-cell and two-cell models previously presented by Malatinszky et al. [36]. The single-cell models properly predicted mixotrophic growth on various carbon sources, however, the two-cell models overestimate the growth under autotrophic and diazotrophic conditions. Hence, gene expression data of vegetative cells and heterocysts for *A. variabilis* ATCC 29413 was used and the regulated two-cell model was constructed. In accordance with previous research, this model predicted that PSII controls growth under high irradiance and heterocyst is a more suitable host for hydrogen in the photoautotrophic and

diazotrophic conditions. The model was then used to predict up/down-regulation strategies to enhance the production of hydrogen. For this purpose, the flux distribution under the minimal and maximal hydrogen production was compared and the reactions whose fluxes increased/ decreased were identified. The proposed changes for the improvement of hydrogen production were in agreement with previous experimental works. The proposed metabolic model not only is useful for in silico study the metabolism of this filamentous, diazotrophic cyanobacterium but also it could be a valuable tool for improvement of bioproducts production using the two-cell system of *A. variabilis*.

## Supporting information

**S1 File. The metabolite and reaction lists of *Anabaena variabilis* ATCC29413 metabolic model and reference papers used for extracting reactions.**
(XLS)

**S2 File. Details of calculations for biomass reaction formula.**
(XLSX)

**S3 File. The mat-files of the two-cell models for implementation in MATLAB.**
(ZIP)

**S4 File. Supplemental figures and tables.**
(DOC)

**S5 File. The list of up- and down-regulated reactions and their fluxes during maximal and minimal hydrogen production.**
(XLS)

## Author Contributions

**Conceptualization:** Ehsan Motamedian.

**Data curation:** Ali Malek Shahkouhi, Ehsan Motamedian.

**Formal analysis:** Ali Malek Shahkouhi, Ehsan Motamedian.

**Investigation:** Ali Malek Shahkouhi, Ehsan Motamedian.

**Methodology:** Ali Malek Shahkouhi, Ehsan Motamedian.

**Project administration:** Ehsan Motamedian.

**Resources:** Ali Malek Shahkouhi.

**Software:** Ehsan Motamedian.

**Supervision:** Ehsan Motamedian.

**Validation:** Ali Malek Shahkouhi, Ehsan Motamedian.

**Writing – original draft:** Ali Malek Shahkouhi, Ehsan Motamedian.

**Writing – review & editing:** Ali Malek Shahkouhi, Ehsan Motamedian.

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
