## [Decision Letter · Decision Letter 0]

26 Sep 2019

PONE-D-19-22664

Reconstruction of a two-cell metabolic model to study biohydrogen production in a diazotrophic cyanobacterium Anabaena variabilis ATCC 29413

PLOS ONE

Dear Dr Ehsan Motamedian,

Thank you for submitting your manuscript to PLOS ONE. After careful consideration by four experts in the field, we feel that it has merit but does not fully meet PLOS ONE’s publication criteria as it currently stands. Therefore, we invite you to submit a revised version of the manuscript that addresses the vraious points raised during the review process. English langage should be improved throughout the manuscript and the manuscript should include a discussion or conclusion section. Furthermore, your model should be compared to other existing models

We would appreciate receiving your revised manuscript by end of october. To enhance the reproducibility of your results, we recommend that if applicable you deposit your laboratory protocols in protocols.io, where a protocol can be assigned its own identifier (DOI) such that it can be cited independently in the future. For instructions see: http://journals.plos.org/plosone/s/submission-guidelines#loc-laboratory-protocols

We look forward to receiving your revised manuscript.

Kind regards,

Marie-Joelle Virolle, PhD

Academic Editor

PLOS ONE

Journal Requirements:

Reviewers' comments:

Reviewer's Responses to Questions

**Comments to the Author**

1. Is the manuscript technically sound, and do the data support the conclusions?

Reviewer #1: Partly

Reviewer #2: Yes

Reviewer #3: Yes

Reviewer #4: Yes

2. Has the statistical analysis been performed appropriately and rigorously? 

Reviewer #1: N/A

Reviewer #2: I Don't Know

Reviewer #3: Yes

Reviewer #4: I Don't Know

3. Have the authors made all data underlying the findings in their manuscript fully available?

Reviewer #1: Yes

Reviewer #2: Yes

Reviewer #3: Yes

Reviewer #4: Yes

4. Is the manuscript presented in an intelligible fashion and written in standard English?

Reviewer #1: Yes

Reviewer #2: Yes

Reviewer #3: Yes

Reviewer #4: No

5. Review Comments to the Author

Reviewer #1: The manuscript entitle “Reconstruction of a two-cell metabolic model to study biohydrogen production in a diazotrophic cyanobacterium Anabaena variabilis ATCC 29413” by Shahkouhi and Motamedian presents modeling work towards the understanding of the phototrophic metabolism of A. variabilis. Authors reconstructed a combined model using two physiological stages of A. variabilis, while studying hydrogen production. Overall, the study is not novel from the modeling perspective because there are available more comprehensive models of Anabaena. Previous studies utilized the same combined model approach, thus, authors should compare their results with modeling outcomes of previously reconstructed models. Additionally, authors should contextualize their experimental evidence for a better data description of the Anabaenas’ metabolism at genome-scale.

Critical issues

The introductory paragraph must highlight, why authors pick the Anabaena variabilis ATCC 29413 strain.

Supplementary file S2 couldn’t be simulated due to problems that have been encountered in the model structure inconsistent Fields: S: mets: Size of S does not match elements in mets, b: mets: Size of b does not match elements in mets. Please provide the Supplementary file 2 again.

Model structure should be compared with other available models for A. variabilis.

Overall, the article is not publishable in this incomplete condition. Authors performed a limited number of analyses, which almost seem like preliminary tests. Moreover, the predictions in this article are not compared or validated with experimental data, which make a confusing situation for the readers. Mostly, the authors’ claims are based on the predictions, which cannot be believed until verified with some experimental data. I was hoping, at least, the scientific community can take some advantage using the proposed model, but the model is in bad shape which cannot be used or propagate by incorporating more pathways or reactions to understand the broader metabolic behavior of the organism. These models contain non-standard reaction and metabolite ids, which do not match to any reaction databases.

Minor general issues

Line 12. Please, correct the organism name

Line 19 to 21. Please reword this sentence

Line 25 to 26. Homogenize capital letters

Line 35. Rephrase as "basic requirements for life are carbon, oxygen and nitrogen"

Line 37, the word "only" is too vague. What is it referring to? Number of different nutrients? Concentrations? Authors should be more specific about this claim

Line 39 to 40, this sentence is confusing

Line 67. Change “sequenced data” by genomic sequenced data.

Line 71. Update references some suggestions are doi: 10.1186/s13068-018-1244-3 and 10.1042/BST20170242.

Line 100. Change “applied” by “used”

Line 142. Delete “Upper and lower limits of all reversible reactions were set to 1000 and -1000 mmol/gDCW/h, respectively. Fluxes were limited between 0 and 1000 mmol/gDCW/h for all intracellular irreversible reactions”.

Figures

Figure 1b – no information is provided about what two different colors are meant for. Figure 2 – this figure is confusing. Irradiance values are missing from some of the small squares. Figure 3 – Panel names (a, b ?) are missing. It is not known what the colors in these heatmaps stand for. What data have been plotted here, which is leveled by the color bar? Figure 4 – these proposed alternations are not beneficial until they are not compared with the experimental data.

Reviewer #2: In this study the authors have built a genome scale metabolic model to analyze the hydrogen production in the heterocystous cyanobacteria Anabaena Variabilis. Both a single cell model and a two-cell model, accounting for the vegetative cell and heterocyst, were constructed. The models were evaluated by comparing their growth rate against experimentally reported values. The two-cell model was first used to compare the oxygen production rate in the vegetative cell and the heterocyst. The analysis showed that the oxygen production is negligible in the heterocyst as compared to vegetative cell and therefore heterocyst is more suitable for hydrogen production. The model was then used to predict up/down-regulation strategies to enhance the production of hydrogen. For this purpose, the flux distribution under the minimal and maximal hydrogen production were compared and the reactions whose fluxes increased/decreased were identified. The manuscript is a nice contribution of a heterocystous cyanobacterium, however, a number of issues need to be resolved first

Major comments:

• Page 6 and line 123-132: What is the source of information about the relative fraction of different lipids (MGDG, SQDG, DGDG and PG)? The author should cite the appropriate reference. Similarly, the text should directly indicate what is the source of information for the glycogen content and pigment content. If the authors have taken the peptidoglycan and LPS content from Synechocystis 6803 model (Nogales et al., 2012) and this needs to be mentioned in the text.

• Section 2.3.1: What is the purpose of the single cell model? This is not made clear in this study. The only place where the single cell model is used in this study is in Section 3.2. There the authors compare the predictions of the single cell model and two-cell model with the experimental data. However, there is no clear discussion about the purpose of using the single cell model in this analysis. The authors should clarify why the analysis using single model is needed for this study.

• Section 2.3.2: Does the two-cell model have separate biomass formation equations for the heterocyst and vegetative cell? If yes, then do the authors assume that their biomass compositions are same/different? The authors should modify the text to include these details.

• Section 2.3.2: What is the biomass objective function for the two-cell model? Is it only the growth of the vegetative cell or the growth of both the cell types in the ratio they are found in the filaments. This a very important information and the authors should incorporate this detail into the text in section 2.3.2

• Section 2.3.2: In Anabaena sp. PCC 7120 model (Malatinszky et al., 2017), the intercellular cellular exchange flux of glutamine and glutamate was fixed to 1. If they are not fixed at this ratio, then the heterocyst super-compartment uses glutamate as carbon source instead of sucrose. What are the constraints on the intercellular exchange fluxes in this study? Did the authors constrain the ratio of intercellular exchange fluxes of glutamate and glutamine? Is sucrose the only carbon source used by the heterocyst compartment or does it utilize glutamate as well? The authors should provide these details in the text.

• One of the major contributions of this work is the model based prediction of up/down regulation strategies for the enhanced hydrogen production. This was accomplished by comparing the FBA predicted flux under minimal and maximal hydrogen production conditions to identify the reactions/pathways that needs to be up/downregulated for the enhanced production of hydrogen. This strategy is not complete since FBA does not always have a unique solution. The more appropriate way is to consider the feasible flux ranges predicted using Flux variability analysis (FVA) (Mahadevan and Schilling, 2003) under the minimal and maximal hydrogen production e.g. (Englund et al., 2018).

Minor comments:

• Page 4 and line 58-61: This sentence is grammatically incorrect. The authors should drop the word ‘since’ in the beginning of the sentence or the ‘and’ in the middle of the sentence.

• Page 4 and line 73-74: Instead of ‘exchange effect of intercellular metabolites’, it is better to use ‘effect of intercellular exchange of metabolites’.

• Page 7 and line 144 to 148: This sentence has grammatical error and that makes it difficult to understand. The others should reframe it or break it into multiple sentences so that it is easy to understand.

• Page 13 and line 298 to 300: The short form for bidirectional hydrogenase is ‘Hox’ not ‘Hop’.

• In some places the authors use ‘Hup’ to refer to the uptake hydrogenase. But in other places they refer to it as ‘uptake hydrogenase’. The authors should use a consistent nomenclature.

• In some places the authors use ‘Hox’ to refer to the bidirectional hydrogenase. But in other places they refer to it as ‘bidirectional hydrogenase’. The authors should use a consistent nomenclature.

• Page 13 and line 296: It is not appropriate to use the word ‘activation’ in this sentence as it conveys a different meaning and the oxidative pentose phosphate pathway is already active under the non-hydrogen-producing condition. The word ‘Utilization’ is more appropriate.

• Figure 3: Which parameter do the pixel color in heat map correspond to. Growth rate? This needs to be mentioned in the figure legend.

References:

1. Englund E, Shabestary K, Hudson EP, Lindberg P (2018) Systematic overexpression study to find target enzymes enhancing production of terpenes in Synechocystis PCC 6803, using isoprene as a model compound. Metabolic Engineering 49: 164-177

2. Mahadevan R, Schilling CH (2003) The effects of alternate optimal solutions in constraint-based genome-scale metabolic models. Metabolic Engineering 5: 264-276

3. Malatinszky D, Steuer R, Jones PR (2017) A Comprehensively Curated Genome-Scale Two-Cell Model for the Heterocystous Cyanobacterium Anabaena sp. PCC 7120. Plant Physiol 173: 509-523

4. Nogales J, Gudmundsson S, Knight EM, Palsson BO, Thiele I (2012) Detailing the optimality of photosynthesis in cyanobacteria through systems biology analysis. Proceedings of the National Academy of Sciences 109: 2678

Reviewer #3: General remarks:

This article discuss the metabolic interactions of the two differentiated cells and construct a two-cell metabolic model to improve hydrogen production. The authors find that the removal of uptake hydrogenase and activation of some reactions to provide redox cofactors by bidirectional hydrogenase could improve hydrogen production. Overall, this article is well organized and its presentation is good. However, some minor issues still need to be improved:

(1) In chapter 1. Introduction: The topic is interesting but the purpose of the article is not clearly stated. After revision please, redefine the introduction and key words.

(2) In chapter 2. Material and Methods: It is suggested to revise the section "2.3. In silico simulations". The detailed model simulation (e.g. calculation formulas and so on) needs to be added to the Supporting Information in order to support the description in the text.

(3) It is suggested to add the part of "4. Conclusion" in this article.

(4) In Table1, the decimal values for points should be given uniformity. For example, the percent of non-gene assigned reactions should be 11.0.

(5) In Figure 1(b), please add the title of the coordinate axis x.

(6) The list of references is not in our style. It is close but not completely correct. Please revised before submitting a revision.

Reviewer #4: Overview:---------------

In the manuscript titled "Reconstruction of a two-cell metabolic model to study biohydrogen production in a diazotrophic cyanobacterium Anabaena variabilis ATCC 29413", the authors describe genome-scale network reconstruction (GENRE) and metabolic reconstruction of Anabaena variabilis ATCC29413. Indeed, this will be the first published metabolic model for the species; and thus, by itself is a valuable tool for not only Anabaena variabilis community but also the metabolic modeling community in general. Apart from this authors discussed biohydrogen production in A. variabilis with its distinct cells: vegetative and heterocytes.

However, I find that manuscript is poorly written and has a lot of grammatical errors. I have tried to list many of them from introduction but stopped keeping track after a while because there were so many. The authors should go through the manuscript and make sure that these instances are fixed.

More importantly, the authors need to provide additional simulations and benchmark their models with existing models of other cyanobacteria or other two-cell models. The only benchmark, Figure 2, is quite opaque and not very informative of the model performance. Figure 3 is missing color axis labels. Figure 4 is not clear from captions. Further, the manuscript does not have a discussion or conclusion section.

Overall, keeping the above points in mind, I think the authors need to undergo major revisions of their manuscript. Please find my detailed comments below.

Comments:---------

MAJOR:

193-208:The authors spend considerable amount of text just reiterating whats in the table and figure. The authors should provide a schematic of how various pathways are organized across different compartments. The authors have to realize that these are large-scale models and a simplified schematic detailing the model organization goes a long way for the readers and potential users of their model.

The authors should also discuss how does this model compare to metabolic network reconstructions of other cyanobacteria such as Synechococcus, Synechocystis, etc. Are these differences represented in their model? How does their model compare with the automatically generated model here (https://www.ebi.ac.uk/biomodels-main/BMID000000140898) or of a different two-cell model here (Malatinszky et al., 2017: http://www.plantphysiol.org/content/173/1/509). Please provide these comparisons.

210-211:In figure 2, I think the authors are trying to compare the growth of their models at different lux. If so, their single cell model is an steady-state FBA model, i.e. it is always linear and will always give global optima of the objective function value. This means that direction of slope of model inputs will be same as that of model outputs (i.e. biomass or growth). They sometimes get a lower growth rate at higher light and higher growth rate at lower light (for e.g. single cell model at 14700 and 8000). This could be because of HCO3 value set by the authors on the model. Basically, this plot is super hard to read and interpret the performance of the model. The authors should also show respective carbon/other nutrient uptake values for these simulations. Just by looking at it as is, I am not sure what's going on with the model.

Further, the authors should also explain why they see this output. What is happening in the model when excess light is available? Give us some flux distributions, please explain your results. There are many number of simulations that can be performed to evaluate the model performance and behavior. What is the accuracy of single gene deletion results? Could authors show some of that. As authors of the model, they should perform and discuss such simulations before jumping on the biohydrogen production.

264: The authors are seeing a growth tradeoff in Figure 3(b). What exaplins this growth tradeoff? Also the figure captions need to be written clearly. How is the electron transport chain playing into this? Is the heatmap representing growth or some other flux. Please label the color axes.

There is no conclusion or discussion section for results. The only results authors discuss is the biohydrogen production. The authors should provide a conclusion section. Have they achieved what they set out to achieve? What are still the outstanding questions in biohydrogen production using Anabaena variabilis ATCC29413 that the model couldn't address. What is needed to improve the model?

MINOR:

48-50: The sentence reads a bit odd. Here's a suggestion: "In absence of nitrate or ammonia in the growth media, vegetative cells undergo heterocyst differentiation enabling nitrogen fixation."

55-58: Again, reads a bit odd. "Heterocysts are characterized by having a thick cell wall that limits the entrance of oxygen, deactivated O2-producing photosystem II, and a high respiration rate that scavenges the remaining oxygen."

59-61: Please check the grammar. Suggestion: "...is absent in the heterocysts; and uses sucrose transported from vegetative cells as the dominant form of carbon."

61-63: Suggestion: "...in heterocysts is transported to vegetative cells.."

64-66: Has metabolic modeling been done before on this species? If so, please provide references. If not, provide explanation for why it is interesting. In the previous paragraph you mention intercellular exchange of metabolites. The presentation of your idea comes across as choppy. Please integrate this idea there on why there is a need for metabolic model; and then, carry on in the next paragraph with "Genome-scale metabolic models..."

74-77: Suggestion: "Eventhough genus Anabena has garnered interest for its biohydrogen production during nitrogen fixation [37-39], its metabolic models have not been used to study biohydrogen production.."

79: Please replace "and researches have been..." with "and has been popular candidate for studying biohydrogen production."

81: Once you say, "hereafter A. variabilis" please use that.

78-95: In the final paragraph, avoid references and authors should just talk about what their work in this study. Anything that needs a reference, should have been said before this point.

155: Please remove "two".

195: Please point the reader to the methods section where you discuss these databases and the process of reconstruction.

280: "improvement"? do the authors mean increase, decrease, and compared to what?

282-295:The paragraph is too word and can be shortened to convey the message succinctly and clearly.

6. PLOS authors have the option to publish the peer review history of their article (what does this mean?). If published, this will include your full peer review and any attached files.

Reviewer #1: No

Reviewer #2: No

Reviewer #3: None

Reviewer #4: No

---

## [Author Response · Author response to Decision Letter 0]

11 Nov 2019

Thank you for your valuable comments. We revised the paper according to your suggestions. Here are our answers to your specified questions.

Reviewer #1:

The manuscript entitle “Reconstruction of a two-cell metabolic model to study biohydrogen production in a diazotrophic cyanobacterium Anabaena variabilis ATCC 29413” by Shahkouhi and Motamedian presents modeling work towards the understanding of the phototrophic metabolism of A. variabilis. Authors reconstructed a combined model using two physiological stages of A. variabilis, while studying hydrogen production. Overall, the study is not novel from the modeling perspective because there are available more comprehensive models of Anabaena. Previous studies utilized the same combined model approach, thus, authors should compare their results with modeling outcomes of previously reconstructed models. Additionally, authors should contextualize their experimental evidence for a better data description of the Anabaenas’ metabolism at genome-scale.

Critical issues

The introductory paragraph must highlight, why authors pick the Anabaena variabilis ATCC 29413 strain.

The manuscript was revised according to your comment and our aim was highlighted.

Supplementary file S2 couldn’t be simulated due to problems that have been encountered in the model structure inconsistent Fields: S: mets: Size of S does not match elements in mets, b: mets: Size of b does not match elements in mets. Please provide the Supplementary file 2 again.

Three metabolic models including single-cell, two-cell, and regulated two-cell models were constructed in this research. The single-cell model was used to simulate the growth of vegetative cells on nitrate as a combined nitrogen source. In the presence of nitrate, heterocyst differentiation does not occur, and hence, the reconstructed metabolic model as a single-cell model was applied to predict the growth rate using FBA. This method expresses a metabolic model as a linear programming problem, as in Eq. (1), and defines an objective function (Z).

 (1)

where c is the vector of the objective function coefficients and S is an m×n matrix with rank ≤ m. m=921 is the number of equations derived from the metabolic model using the mass balance for each metabolite and n=983 is the number of unknown reaction fluxes. b is the right-hand side vector determined by known reaction fluxes, and vmin and vmax are the lower and upper bounds of the variable fluxes, respectively. Upper and lower bounds of all intracellular reversible reactions were set to 1000 and -1000 mmol/gDCW/h, respectively. Fluxes were limited between 0 and 1000 mmol/gDCW/h for all intracellular irreversible reactions. Upper and lower bounds of all exchange reactions were set to 1000 mmol/gDCW/h and zero respectively. Lower bound of exchange reactions for H+, H2O, K, SO4, Pi, and inorganic ions was set to -1000 mmol/gDCW/h and lower bound of exchange reactions for HCO3, NO3, photon was set to -10 mmol/gDCW/h. Maximum uptake rate of each carbon source was determined according to the method presented in [1].

The biomass reaction was used as the objective function to be maximized using FBA and the single-cell model was compared with the metabolic model of Anabaena sp. PCC 7120 [1] for prediction of growth on different carbon sources. So, according to the method presented in [1], mixotrophic growth rate for each carbon source was calculated and divided by autotrophic growth rate on bicarbonate. Correlation between the predicted relative growth rates and experimental values for the two models was determined using the Pearson correlation coefficient. 

The two-cell model was constructed to compare its growth predictions under autotrophic and diazotrophic conditions with those predicted by the regulated two-cell model and predicted by the previous two-cell model [1]. Upper and lower bounds of all intracellular reactions for the heterocyst and vegetative metabolic models were the same as those for the single-cell model, except those presented in Table S1 (supplementary file S4). As nitrogenase is inhibited irreversibly by oxygen and concentration of oxygen in vegetative cells is high [2], it was removed from the metabolic model of the vegetative cell. Reactions RBCh, RBPC, HCO3E, GLMS, and PSII are only present in vegetative cells according to the references presented in Table S1. Oxygen evolving photosystem II (PSII) is inactivated in heterocyst [3] because nitrogenase is oxygen-sensitive and in the presence of O2 will be deactivated irreversibly. Besides, the absence of the key enzyme of the reductive pentose phosphate pathway, ribulose 1,5-bisphosphate carboxylase (RBCh and RBPC), in heterocyst has been reported in the literature before [4, 5]. Presence of glutamine-2-oxoglutarate aminotransferase (Fd-GOGAT) which is the producer of glutamate from 2-oxoglutarate and glutamine in heterocysts of A. variabilis was doubtful before [6], but later it was clearly concluded that heterocysts lack this enzyme [7].

Since ribulose 1,5-bisphosphate carboxylase is absent in heterocysts of A. variabilis, fixed carbon in the form of sucrose transfers from nearby vegetative to heterocyst cells as carbon source. In return, heterocysts provide the nitrogen source needed for the metabolism of vegetative cells. Glutamate transports from vegetative cells to heterocysts and the fixed nitrogen in the form of glutamine with an additional nitrogen atom than glutamate with the same carbon skeleton shuttles back to vegetative cells as the nitrogen source for growth [8]. Hence, the entire filament can grow by incorporating fixed nitrogen provided in heterocysts. The exchange of these three metabolites (glutamine, glutamate, and sucrose) between the two cell types happens through a continuous periplasm [9, 10].

Upper and lower bounds of all exchange reactions for the two-cell model were the same as those for the single-cell model, except for HCO3, N2, NO3, and photon. Uptake of HCO3 was only considered for vegetative cells and consumption of N2 was only considered for heterocysts. To simulate the diazotrophic condition, uptake of NO3 was limited to zero. Considering both vegetative and heterocyst cells can uptake photon, Eq. 2 was added to the model. It should be mentioned that a slack variable was added to the right-hand side of Eq. 2 to transform the inequality constraint into equality.

 (2)

vp,max indicates the maximum uptake rate of photon, and vp,V and vp,H are photon uptake rates of vegetative cells and heterocysts, respectively. The maximum uptake rates of HCO3 and photon were set to experimental values reported by [11].

The biomass reaction of the vegetative cells was used as the objective function to be maximized using FBA. The biomass of heterocyst cells is 5-10 percent of vegetative cells biomass [2, 12]. In this research, it was assumed that biomass composition of heterocysts is the same as vegetative cells and heterocysts comprise 10 percent of the entire filament and therefore, Eq. 3 was added to the two-cell model.

 (3)

vg,V and vg,H are growth rates of vegetative and heterocyst cells, respectively. The calculated biomass objective function was multiplied by 1.1 for two-cell models to report the total growth rate. The two-cell model includes 1970 variables including reaction fluxes of the two cells, three transport reactions (for intercellular exchange of sucrose, glutamine, and glutamate), and a slack variable for Eq. 2. This model also includes 1844 equations including 1842 equations from the two metabolic models and Eqs. 2 and 3. Prefixes Vegetative and Heterocyst were used to indicate metabolites and reactions of the vegetative and heterocyst models, respectively. It should be mentioned that Eqs. 2 and 3 were also added to the two-cell model presented by Malatinszky et al. [1]. 

The TRFBA algorithm presented by Motamedian et al. [13] was applied to integrate gene expression data [14] of vegetative and heterocyst cells in photoautotrophic and heterotrophic conditions for A. variabilis ATCC 29413 with the two metabolic models. This algorithm first converted the metabolic model to irreversible and “without OR” form that increased the number of variables to 3669 for the two-cell model. Then, TRFBA added a constraint for each metabolic gene with measured expression level to the two-cell model using Eq. (4).

 (4)

where Kj denotes the set of indices of reactions supported by metabolic gene j, vi is the flux of reaction i, and Ej is the expression level of jth gene. C is a constant parameter that converts the expression levels to the upper bounds of the reactions. The unit for C is mmol gDCW-1h-1 and this coefficient indicates the maximum rate supported by one unit of the expression level of a gene. The optimal value of C was determined using sensitivity analysis. By adding a positive slack variable (aj) for each gene to Eq. 4, the inequality constraint is transformed into an equality constraint (Eq. 5).

 (5)

The expression level of 951 metabolic genes was measured for heterocysts and vegetative cells and hence, 1902 variables and equations were added to the regulated two-cell model. The added equations and variables were named with their corresponding gene and prefix Vegetative or Heterocyst. The maximum uptake rate of HCO3 was set to 1 mmol/gDCW/h and the maximum uptake rate of N2 and photon was set to 1000 mmol/gDCW/h to simulate photoautotroph and diazotrophic growth condition. The maximum uptake rate of HCO3 and photon was set to experimental values [11] for comparison with the two-cell models. To simulate the heterotrophic growth condition using the regulated two-cell model, fructose as carbon and energy source with the maximum uptake rate of 1 mmol/gDCW/h was selected as carbon and energy sources for vegetative cells instead of HCO3 and photon.

Shadow price was also calculated for each metabolic gene to determine the gene that controls the growth rate of A. variabilis in high irradiance. Shadow price shows sensitivity of the objective function to change in the right-hand side vector of Eq. (5) according to Eq. 6.

 (6)

Shadow price indicates the effect of one unit increase in gene expression on the predicted growth rate of A. variabilis. Its positive value for a gene shows that the expression level of the gene is not adequate and this level as an intracellular constraint controls the metabolism for growth.

The name of each added equation was added to model.mets and so size of b match elements in mets. According to the explanations, the added names are not metabolites name and they are name of genes and the selected names for Eqs. 2 and 3. The explanations were added to the manuscript.

Model structure should be compared with other available models for A. variabilis.

A comparison between the metabolic models of Anabaena variabilis ATCC 29413, Synechocystis sp. PCC6803, Synechococcus elongatus PCC 7942, Spirulina platensis, Anabaena sp. PCC 7120 has been made in the Table 1.

Overall, the article is not publishable in this incomplete condition. Authors performed a limited number of analyses, which almost seem like preliminary tests. Moreover, the predictions in this article are not compared or validated with experimental data, which make a confusing situation for the readers. Mostly, the authors’ claims are based on the predictions, which cannot be believed until verified with some experimental data.

More simulations for growth prediction on various carbon sources and investigation of the effect of heterocyst percentage were performed. The predictions of the single-cell and two-cell models presented by Malatinszky et al. [1] were compared with our models for the simulation of mixotrophic and autotrophic growth.

I was hoping, at least, the scientific community can take some advantage using the proposed model, but the model is in bad shape which cannot be used or propagate by incorporating more pathways or reactions to understand the broader metabolic behavior of the organism. These models contain non-standard reaction and metabolite ids, which do not match to any reaction databases.

The abbreviated names of reactions and metabolites were determined according to the standard names presented in BIGG. Prefixes Vegetative and Heterocyst were used to indicate metabolites and reactions of the vegetative and heterocyst models, respectively. As presented in the response to your second comment, we also determined a name for the other added equations and variables in the two-cell models for more clarification.

Minor general issues

Line 12. Please, correct the organism name

The organism name was corrected

Line 19 to 21. Please reword this sentence

The sentence was reworded as: 

"It predicts that heterocysts provide an oxygen-free environment and then, the regulated model was used to find strategies for improvement of hydrogen production in heterocysts."

Line 25 to 26. Homogenize capital letters

Capital letters were homogenized.

Line 35. Rephrase as "basic requirements for life are carbon, oxygen and nitrogen"

The sentence was rephrased as: 

Cyanobacteria play a key role in providing the primary elements for life including organic carbon, oxygen, and nitrogen.

Line 37, the word "only" is too vague. What is it referring to? Number of different nutrients? Concentrations? Authors should be more specific about this claim

The word "only" was omitted.

Line 39 to 40, this sentence is confusing

The sentence " The energy of sunlight in association with CO2 fixation process converts inorganic carbons to organic ones as carbon skeleton of organisms", was omitted from the manuscript. This sentence was not necessary for the text.

Line 67. Change “sequenced data” by genomic sequenced data.

The phrase was changed.

Line 71. Update references some suggestions are doi: 10.1186/s13068-018-1244-3 and 10.1042/BST20170242.

Thanks for your suggestions, but our paper was focused on metabolic models of cyanobacteria. 10.1186/s13068-018-1244-3 is about a eukaryotic microalga, and in the paper 10.1042/BST20170242, no metabolic model has been reconstructed.

Line 100. Change “applied” by “used”

In this line, "applied" was changed to "used".

Line 142. Delete “Upper and lower limits of all reversible reactions were set to 1000 and -1000 mmol/gDCW/h, respectively. Fluxes were limited between 0 and 1000 mmol/gDCW/h for all intracellular irreversible reactions”.

The word “intracellular” was added to make the sentence meaningful.

Figures

Figure 1b – no information is provided about what two different colors are meant for.

The blue color is for gene associated reactions and red color is for non-gene associated reactions. this explanation was added to the figure 1b.

Figure 2 – this figure is confusing. Irradiance values are missing from some of the small squares.

The same colors were used for the distant points but with the same data to make the figure clear.

Figure 3 – Panel names (a, b ?) are missing. It is not known what the colors in these heatmaps stand for. What data have been plotted here, which is leveled by the color bar?

(a) and (b) were added. The heatmap presents changes in the growth rate with varying oxygen exchange and hydrogen production fluxes in a) vegetative and b) heterocyst cells. The figure caption was corrected.

Figure 4 – these proposed alternations are not beneficial until they are not compared with the experimental data.

The proposed changes for the improvement of hydrogen production was in agreement with previous experimental works.

Reviewer #2:

In this study the authors have built a genome scale metabolic model to analyze the hydrogen production in the heterocystous cyanobacteria Anabaena Variabilis. Both a single cell model and a two-cell model, accounting for the vegetative cell and heterocyst, were constructed. The models were evaluated by comparing their growth rate against experimentally reported values. The two-cell model was first used to compare the oxygen production rate in the vegetative cell and the heterocyst. The analysis showed that the oxygen production is negligible in the heterocyst as compared to vegetative cell and therefore heterocyst is more suitable for hydrogen production. The model was then used to predict up/down-regulation strategies to enhance the production of hydrogen. For this purpose, the flux distribution under the minimal and maximal hydrogen production were compared and the reactions whose fluxes increased/decreased were identified. The manuscript is a nice contribution of a heterocystous cyanobacterium, however, a number of issues need to be resolved first

Major comments:

• Page 6 and line 123-132: What is the source of information about the relative fraction of different lipids (MGDG, SQDG, DGDG and PG)? The author should cite the appropriate reference. Similarly, the text should directly indicate what is the source of information for the glycogen content and pigment content. If the authors have taken the peptidoglycan and LPS content from Synechocystis 6803 model (Nogales et al., 2012) and this needs to be mentioned in the text.

The references were added to the manuscript. In addition, the reference of each data in supplementary file 3 was exactly determined.

• Section 2.3.1: What is the purpose of the single cell model? This is not made clear in this study. The only place where the single cell model is used in this study is in Section 3.2. There the authors compare the predictions of the single cell model and two-cell model with the experimental data. However, there is no clear discussion about the purpose of using the single cell model in this analysis. The authors should clarify why the analysis using single model is needed for this study.

In this first version of the manuscript, we used the single-cell model to indicate the effect of regulation of metabolic model by integration of transcriptomic data but it was not a fair comparison. So two-cell model was constructed to compare its growth predictions under auto- and diazo-trophic condition with those predicted by the regulated two-cell model and predicted by the previous two-cell model [1]. Then, the single-cell model was used to simulate the growth of vegetative cells on nitrate as a combined nitrogen source to evaluate its prediction for growth on different carbon sources compared to the previous metabolic model of Anabaena [1]. In the presence of nitrate, heterocyst differentiation does not occur, and hence, the reconstructed metabolic model as a single-cell model was applied to predict the growth rate using FBA. According to the method presented in [1], mixotrophic growth rate for each carbon source was calculated and divided by autotrophic growth rate on bicarbonate. Correlation between the predicted relative growth rates for the two models and experimental values was determined using the Pearson correlation coefficient.

The manuscript was modified to indicate the changes.

• Section 2.3.2: Does the two-cell model have separate biomass formation equations for the heterocyst and vegetative cell? If yes, then do the authors assume that their biomass compositions are same/different? The authors should modify the text to include these details.

Yes. In this research, it was assumed that the biomass composition of heterocysts is the same as vegetative cells. The assumption was added to the manuscript.

• Section 2.3.2: What is the biomass objective function for the two-cell model? Is it only the growth of the vegetative cell or the growth of both the cell types in the ratio they are found in the filaments. This a very important information and the authors should incorporate this detail into the text in section 2.3.2

The biomass reaction of the vegetative cell was used as the objective function to be maximized using FBA. The biomass of heterocyst cells is 5-10 percent of vegetative cells biomass [2, 12] and so it was assumed that heterocysts comprise 10 percent of the entire filament and Eq. 3 was added to the two-cell models.

 (3)

vg,V and vg,H are growth rates of vegetative and heterocyst cells, respectively. The calculated biomass objective function was multiplied by 1.1 for two-cell models to report the total growth rate.

The information were added to the manuscript.

• Section 2.3.2: In Anabaena sp. PCC 7120 model (Malatinszky et al., 2017), the intercellular cellular exchange flux of glutamine and glutamate was fixed to 1. If they are not fixed at this ratio, then the heterocyst super-compartment uses glutamate as carbon source instead of sucrose. What are the constraints on the intercellular exchange fluxes in this study? Did the authors constrain the ratio of intercellular exchange fluxes of glutamate and glutamine? Is sucrose the only carbon source used by the heterocyst compartment or does it utilize glutamate as well? The authors should provide these details in the text.

We did not impose any constraint on the intercellular exchange fluxes in this study and the three two-cell models were used to predict the fluxes. For a fair comparison between sucrose and glutamate, the rates were converted to mmol carbon/gDCW/h and the predicted exchange rates of sucrose and glutamate were multiplied by 12 and 5, respectively. In addition, FVA [15] was applied to find flux variability of each intercellular exchange flux. The two-cell model of Malatinszky et al. [1] predicts that both sucrose and glutamate can be used as the main carbon source (Figure S3 (a)). Hence, they fixed the glutamine-glutamate exchange ratio to one and forced the model to select sucrose as main carbon source. However, our two-cell model predicts that sucrose is transported to heterocysts as carbon source in all conditions (Figure S3 (b)). Experimental reports also suggest that the main carbon source for heterocysts is sucrose [16]. The regulated two-cell model also transports sucrose to heterocyst in low irradiance (Figure S3 (c)). But both substrates can be used as the main carbon source at high irradiance. This result is rational because the model predicts that the growth rate is constant at high irradiance because of electron limitation and given the extra bicarbonate available, none of the sources are preferred for carbon transfer.

The explanations were added to the manuscript.

• One of the major contributions of this work is the model based prediction of up/down regulation strategies for the enhanced hydrogen production. This was accomplished by comparing the FBA predicted flux under minimal and maximal hydrogen production conditions to identify the reactions/pathways that needs to be up/downregulated for the enhanced production of hydrogen. This strategy is not complete since FBA does not always have a unique solution. The more appropriate way is to consider the feasible flux ranges predicted using Flux variability analysis (FVA) (Mahadevan and Schilling, 2003) under the minimal and maximal hydrogen production e.g. (Englund et al., 2018).

In this research, we used TRFBA for the prediction of up/down regulation strategies. Considering TRFBA converts a metabolic model to “without OR” form, it is possible to have more than one reaction with the same formula that are supported by different genes. So, the application of FVA did not guide us to make a fair decision about the effect of each reaction on hydrogen production. To avoid the well-known degeneracy of solutions, the Manhattan norm of the flux distribution was minimized while the growth and hydrogen production rates were bound to the intended values (similar to the method used for our previous publication [17]).

The sentence “To avoid the well-known degeneracy of solutions, the Manhattan norm of the flux distribution was minimized while the growth and hydrogen production rates were bound to the intended values.” was added to the manuscript for more clarification.

Minor comments:

• Page 4 and line 58-61: This sentence is grammatically incorrect. The authors should drop the word ‘since’ in the beginning of the sentence or the ‘and’ in the middle of the sentence.

According to your comment, the sentence was corrected grammatically as "Since Ribulose 1,5-bisphosphate (RuBP carboxylase, a key enzyme in fixing carbon dioxide) is absent in the heterocysts, sucrose is transported from vegetative cells to heterocysts as the dominant form of carbon."

• Page 4 and line 73-74: Instead of ‘exchange effect of intercellular metabolites’, it is better to use ‘effect of intercellular exchange of metabolites’.

“exchange effect of intercellular metabolites” was substituted with “effect of intercellular exchange of metabolites”.

• Page 7 and line 144 to 148: This sentence has grammatical error and that makes it difficult to understand. The others should reframe it or break it into multiple sentences so that it is easy to understand.

The sentence was corrected.

• Page 13 and line 298 to 300: The short form for bidirectional hydrogenase is ‘Hox’ not ‘Hop’.

The word was corrected.

• In some places the authors use ‘Hup’ to refer to the uptake hydrogenase. But in other places they refer to it as ‘uptake hydrogenase’. The authors should use a consistent nomenclature.

The manuscript was modified.

• In some places the authors use ‘Hox’ to refer to the bidirectional hydrogenase. But in other places they refer to it as ‘bidirectional hydrogenase’. The authors should use a consistent nomenclature.

The manuscript was modified.

• Page 13 and line 296: It is not appropriate to use the word ‘activation’ in this sentence as it conveys a different meaning and the oxidative pentose phosphate pathway is already active under the non-hydrogen-producing condition. The word ‘Utilization’ is more appropriate.

Utilization was substituted.

• Figure 3: Which parameter do the pixel color in heat map correspond to. Growth rate? This needs to be mentioned in the figure legend.

Yes, it is growth rate. The figure legend was modified by adding the sentence “The heatmap presents changes in the growth rate with varying oxygen exchange and hydrogen production fluxes in a) vegetative and b) heterocyst cells.”

References:

1. Englund E, Shabestary K, Hudson EP, Lindberg P (2018) Systematic overexpression study to find target enzymes enhancing production of terpenes in Synechocystis PCC 6803, using isoprene as a model compound. Metabolic Engineering 49: 164-177

2. Mahadevan R, Schilling CH (2003) The effects of alternate optimal solutions in constraint-based genome-scale metabolic models. Metabolic Engineering 5: 264-276

3. Malatinszky D, Steuer R, Jones PR (2017) A Comprehensively Curated Genome-Scale Two-Cell Model for the Heterocystous Cyanobacterium Anabaena sp. PCC 7120. Plant Physiol 173: 509-523

4. Nogales J, Gudmundsson S, Knight EM, Palsson BO, Thiele I (2012) Detailing the optimality of photosynthesis in cyanobacteria through systems biology analysis. Proceedings of the National Academy of Sciences 109: 2678

Reviewer #3:

General remarks:

This article discuss the metabolic interactions of the two differentiated cells and construct a two-cell metabolic model to improve hydrogen production. The authors find that the removal of uptake hydrogenase and activation of some reactions to provide redox cofactors by bidirectional hydrogenase could improve hydrogen production. Overall, this article is well organized and its presentation is good. However, some minor issues still need to be improved:

(1) In chapter 1. Introduction: The topic is interesting but the purpose of the article is not clearly stated. After revision please, redefine the introduction and key words.

According to your comment, the introduction and key words were redefined. The purpose of selecting this organism to build a metabolic model was stated clearly.

(2) In chapter 2. Material and Methods: It is suggested to revise the section "2.3. In silico simulations". The detailed model simulation (e.g. calculation formulas and so on) needs to be added to the Supporting Information in order to support the description in the text.

The section was completely revised and all calculations and formulas were clearly explained.

(3) It is suggested to add the part of "4. Conclusion" in this article.

A conclusion section was added to the manuscript. 

(4) In Table1, the decimal values for points should be given uniformity. For example, the percent of non-gene assigned reactions should be 11.0.

Table 1 was modified.

(5) In Figure 1(b), please add the title of the coordinate axis x.

The title “number of non-gene associated (red) and gene associated (blue) reactions in each category.” was added to the figure legend.

(6) The list of references is not in our style. It is close but not completely correct. Please revised before submitting a revision.

The style of references was corrected.

Reviewer #4:

Overview:---------------

In the manuscript titled "Reconstruction of a two-cell metabolic model to study biohydrogen production in a diazotrophic cyanobacterium Anabaena variabilis ATCC 29413", the authors describe genome-scale network reconstruction (GENRE) and metabolic reconstruction of Anabaena variabilis ATCC29413. Indeed, this will be the first published metabolic model for the species; and thus, by itself is a valuable tool for not only Anabaena variabilis community but also the metabolic modeling community in general. Apart from this authors discussed biohydrogen production in A. variabilis with its distinct cells: vegetative and heterocytes.

However, I find that manuscript is poorly written and has a lot of grammatical errors. I have tried to list many of them from introduction but stopped keeping track after a while because there were so many. The authors should go through the manuscript and make sure that these instances are fixed.

The manuscript was revised grammatically.

More importantly, the authors need to provide additional simulations and benchmark their models with existing models of other cyanobacteria or other two-cell models. The only benchmark, Figure 2, is quite opaque and not very informative of the model performance. Figure 3 is missing color axis labels. Figure 4 is not clear from captions. Further, the manuscript does not have a discussion or conclusion section.

According to your comments, more explanations, simulations, and evaluations were provided. Figures were modified and a conclusion section was added.

Overall, keeping the above points in mind, I think the authors need to undergo major revisions of their manuscript. Please find my detailed comments below.

Comments:---------

MAJOR:

193-208:The authors spend considerable amount of text just reiterating whats in the table and figure. The authors should provide a schematic of how various pathways are organized across different compartments. The authors have to realize that these are large-scale models and a simplified schematic detailing the model organization goes a long way for the readers and potential users of their model.

A simplified schematic of the heterocyst and vegetative cell was drawn in Figure 4 of the manuscript. In Figure 4, the major pathways proposed by the regulated two-cell model for improvement of hydrogen production are presented.

The authors should also discuss how does this model compare to metabolic network reconstructions of other cyanobacteria such as Synechococcus, Synechocystis, etc. Are these differences represented in their model? How does their model compare with the automatically generated model here (https://www.ebi.ac.uk/biomodels-main/BMID000000140898) or of a different two-cell model here (Malatinszky et al., 2017: http://www.plantphysiol.org/content/173/1/509). Please provide these comparisons.

A comparison between the metabolic models of Anabaena variabilis ATCC 29413, Synechocystis sp. PCC6803, Synechococcus elongatus PCC 7942, Spirulina platensis, Anabaena sp. PCC 7120 has been made in the Table 1.

The automatically generated model is a weak model that is reconstructed based on genome annotation. It is a draft model that can not produce biomass and photon uptake is not considered in it. In addition to the genome annotation, we used a lot of information presented in the literature for the model reconstruction and biomass generation. 160 reference papers used for extracting reactions that were presented in supplementary file S1. Mass and charge balances were carried out on each reaction. The used papers for generation of biomass formula are presented in supplementary file S3. The intercellular exchange reactions were added based on literature. We think that automatic reconstruction, especially for Anabaena variabilis that differentiates under nitrogen deficiency condition, is not a suitable approach to construct metabolic models.

The predictions of the reconstructed single-cell and two-cell models in this research were compared with those presented by the single-cell and two-cell models presented by Malatinszky et al. [1]. The results were added to the manuscript.

210-211:In figure 2, I think the authors are trying to compare the growth of their models at different lux. If so, their single cell model is an steady-state FBA model, i.e. it is always linear and will always give global optima of the objective function value. This means that direction of slope of model inputs will be same as that of model outputs (i.e. biomass or growth). They sometimes get a lower growth rate at higher light and higher growth rate at lower light (for e.g. single cell model at 14700 and 8000). This could be because of HCO3 value set by the authors on the model. Basically, this plot is super hard to read and interpret the performance of the model. The authors should also show respective carbon/other nutrient uptake values for these simulations. Just by looking at it as is, I am not sure what's going on with the model. Further, the authors should also explain why they see this output. What is happening in the model when excess light is available? Give us some flux distributions, please explain your results.

We added the HCO3 values to Figure 2 for better readability and interpretation and we also calculated the shadow price to find the bottleneck gene and to explain our results. Both two-cell models overestimate the growth rate under high irradiances and bicarbonate uptake rates. FBA does not generally apply any intracellular constraint and only extracellular constraints including bicarbonate and photon uptake rates control the predicted growth rates. It can be seen that the high rates of bicarbonate consumption belong to high irradiance conditions and the limiting step should be determined. Overprediction for high irradiances and uptake rates indicates that intracellular constraints are inhibiting the growth and so, expression data were integrated with the two-cell metabolic model using TRFBA to impose the intracellular constraints and to find the origin of the inhibition. TRFBA limits the upper bound of intracellular reactions based on the expression level of their corresponding genes. The calculated shadow prices for metabolic genes indicate that the growth rate is sensitive to the expression level of Ava_1852 in the vegetative cells. This gene is responsible for synthesis of protein PsbJ that is one of the components of the core complex of photosystem II. The mean expression level of Ava_1852 is 56 that is significantly less than the average expression level. Hence, TRFBA predicts that PSII is limiting reaction which controls growth in high levels of radiation and bicarbonate uptake. The inhibited growth of A. variabilis at high irradiance has also been stated by Yoon et al. [18, 19]. It is also expressed that high light intensities destroy photosystem II reaction centers and lead to photoinhibition [20, 21].

The explanations were added to the manuscript.

There are many number of simulations that can be performed to evaluate the model performance and behavior. What is the accuracy of single gene deletion results? Could authors show some of that. As authors of the model, they should perform and discuss such simulations before jumping on the biohydrogen production.

More simulations for growth prediction on various carbon sources and investigation of the effect of heterocyst percentage were performed. The predictions of the single-cell and two-cell models presented by Malatinszky et al. [1] were compared with our models for the simulation of mixotrophic and autotrophic growth.

The result of an experimental work [22] that showed more hydrogen is produced in a medium including higher heterocyst frequency was mentioned in the section “Selection of the appropriate cell for hydrogen production”.

Furthermore, the effect of deletion of Hup and Hox on hydrogen production was in accordance with the model prediction that is presented in the section “improvement of hydrogen production”.

264: The authors are seeing a growth tradeoff in Figure 3(b). What exaplins this growth tradeoff? Also the figure captions need to be written clearly. How is the electron transport chain playing into this? Is the heatmap representing growth or some other flux. Please label the color axes.

The colors and color bars were corrected. The figure caption was modified. The electron produced from electron transport chain can be used for growth or hydrogen production. So, double robustness analysis of PSII and hydrogen production rate is similar to that is plotted for oxygen and hydrogen production rates in Figure 3 (a).

There is no conclusion or discussion section for results. The only results authors discuss is the biohydrogen production. The authors should provide a conclusion section. Have they achieved what they set out to achieve? What are still the outstanding questions in biohydrogen production using Anabaena variabilis ATCC29413 that the model couldn't address. What is needed to improve the model?

The conclusion section was added to the manuscript.

MINOR:

48-50: The sentence reads a bit odd. Here's a suggestion: "In absence of nitrate or ammonia in the growth media, vegetative cells undergo heterocyst differentiation enabling nitrogen fixation."

According to your comment, the sentence in lines 48-50 was substituted with: "In absence of nitrate or ammonia in the growth media, vegetative cells undergo heterocyst differentiation enabling nitrogen fixation." 

55-58: Again, reads a bit odd. "Heterocysts are characterized by having a thick cell wall that limits the entrance of oxygen, deactivated O2-producing photosystem II, and a high respiration rate that scavenges the remaining oxygen."

We rewrtie the sentence as yours: "Heterocysts are characterized by having a thick cell wall that limits the entrance of oxygen, deactivated O2-producing photosystem II, and a high respiration rate that scavenges the remaining oxygen."

59-61: Please check the grammar. Suggestion: "...is absent in the heterocysts; and uses sucrose transported from vegetative cells as the dominant form of carbon."

The sentece was corrected.

61-63: Suggestion: "...in heterocysts is transported to vegetative cells.."

The sentence in lines 61-63 was substituted with:"In return, fixed nitrogen in heterocysts is transported to vegetative cells as amino acids and hence, the entire filament will grow by the intercellular exchange of metabolites."

64-66: Has metabolic modeling been done before on this species? If so, please provide references. If not, provide explanation for why it is interesting. In the previous paragraph you mention intercellular exchange of metabolites. The presentation of your idea comes across as choppy. Please integrate this idea there on why there is a need for metabolic model; and then, carry on in the next paragraph with "Genome-scale metabolic models..."

The sentences were changed to indicate our idea: “Even though genus Anabena has garnered interest for its biohydrogen production during nitrogen fixation [23-25], its metabolic models have not been used to study biohydrogen production. Anabaena variabilis ATCC 29413 (hereafter A. variabilis) has one of the highest hydrogen production rates in cyanobacteria and has been popular candidate for studying biohydrogen production [26, 27]. Therefore, a curated genome-scale metabolic model for A. variabilis ATCC 29413 was reconstructed in this research for the first time to study the metabolism of biohydrogen production.”

74-77: Suggestion: "Eventhough genus Anabena has garnered interest for its biohydrogen production during nitrogen fixation [37-39], its metabolic models have not been used to study biohydrogen production.."

According to your comment, lines 74-77 were rewritten as: "Even though genus Anabena has garnered interest for its biohydrogen production during nitrogen fixation [37-39], its metabolic models have not been used to study biohydrogen production."

79: Please replace "and researches have been..." with "and has been popular candidate for studying biohydrogen production."

The sentence was corrected based on your comment.

81: Once you say, "hereafter A. variabilis" please use that.

The abbreviation was used in the manuscript.

78-95: In the final paragraph, avoid references and authors should just talk about what their work in this study. Anything that needs a reference, should have been said before this point.

The last paragraph was seperated from the other parts of the introduction.

155: Please remove "two".

The word “two” was removed.

195: Please point the reader to the methods section where you discuss these databases and the process of reconstruction.

The sentence: "The reconstruction process has been discussed in the material and methods section." was added according to your comment.

280: "improvement"? do the authors mean increase, decrease, and compared to what?

The word "imrpovement" was substituted with "increasing the" according to your comment. the model needs to have more sucrose to be transported from the vegetative cells to heterocysts to increase the growth rate. 

282-295:The paragraph is too word and can be shortened to convey the message succinctly and clearly.

Yes, the paragraph was too word and it was shortened in two paragraphs. 

References

1. Malatinszky D, Steuer R, Jones PR. A comprehensively curated genome-scale two-cell model for the heterocystous cyanobacterium Anabaena sp. PCC 7120. Plant physiology. 2017;173(1):509-23.

2. Adams DG. Heterocyst formation in cyanobacteria. Current opinion in microbiology. 2000;3(6):618-24.

3. Donze M, Haveman J, Schiereck P. Absence of photosystem 2 in heterocysts of blue-green alga Anabaena. Biochim Biophys Acta. 1972;256.

4. Curatti L, Flores E, Salerno G. Sucrose is involved in the diazotrophic metabolism of the heterocyst-forming cyanobacterium Anabaena sp. FEBS Lett. 2002;513.

5. Kolman MA, Nishi CN, Perez-Cenci M, Salerno GL. Sucrose in cyanobacteria: from a salt-response molecule to play a key role in nitrogen fixation. Life. 2015;5(1):102-26.

6. Rai AN, Rowell P, Stewart WD. Glutamate Synthase Activity of Heterocysts and Vegetative Cells of the Cyanobacterium Anabaena variabilis Kütz. Microbiology. 1982;128(9):2203-5.

7. Martín-Figueroa E, Navarro F, Florencio FJ. The GS-GOGAT pathway is not operative in the heterocysts. Cloning and expression of glsF gene from the cyanobacterium Anabaena sp. PCC 7120. FEBS Lett. 2000;476.

8. Thomas J, Meeks JC, Wolk CP, Shaffer PW, Austin SM. Formation of glutamine from 13N-ammonia, 13N-dinitrogen, and 14C-glutamate by heterocysts isolated from Anabaena cylindrica. J Bacteriol. 1977;129.

9. Flores E, Herrero A. Compartmentalized function through cell differentiation in filamentous cyanobacteria. Nat Rev Microbiol. 2010;8.

10. Mariscal V, Herrero A, Flores E. Continuous periplasm in a filamentous, heterocyst‐forming cyanobacterium. Molecular microbiology. 2007;65(4):1139-45.

11. Berberoğlu H, Barra N, Pilon L, Jay J. Growth, CO₂ consumption and H₂ production of Anabaena variabilis ATCC 29413-U under different irradiances and CO₂ concentrations. Journal of applied microbiology. 2008.

12. Kumar K, Mella-Herrera RA, Golden JW. Cyanobacterial heterocysts. Cold Spring Harb Perspect Biol. 2009;2.

13. Motamedian E, Mohammadi M, Shojaosadati SA, Heydari M. TRFBA: an algorithm to integrate genome-scale metabolic and transcriptional regulatory networks with incorporation of expression data. Bioinformatics. 2016;33(7):1057-63.

14. Park J-J, Lechno-Yossef S, Wolk CP, Vieille C. Cell-specific gene expression in Anabaena variabilis grown phototrophically, mixotrophically, and heterotrophically. BMC genomics. 2013;14(1):759.

15. Mahadevan R, Schilling C. The effects of alternate optimal solutions in constraint-based genome-scale metabolic models. Metabolic engineering. 2003;5(4):264-76.

16. Nürnberg DJ, Mariscal V, Bornikoel J, Nieves-Morión M, Krauß N, Herrero A, et al. Intercellular diffusion of a fluorescent sucrose analog via the septal junctions in a filamentous cyanobacterium. MBio. 2015;6(2):e02109-14.

17. Motamedian E, Saeidi M, Shojaosadati S. Reconstruction of a charge balanced genome-scale metabolic model to study the energy-uncoupled growth of Zymomonas mobilis ZM1. Molecular BioSystems. 2016;12(4):1241-9.

18. Yoon JH, Sim SJ, Kim M-S, Park TH. High cell density culture of Anabaena variabilis using repeated injections of carbon dioxide for the production of hydrogen. International journal of hydrogen energy. 2002;27(11-12):1265-70.

19. Yoon JH, Shin J-H, Park TH. Characterization of factors influencing the growth of Anabaena variabilis in a bubble column reactor. Bioresource technology. 2008;99(5):1204-10.

20. Agel G, Nultsch W, Rhiel E. Photoinhibition and its wavelength dependence in the cyanobacterium Anabaena variabilis. Archives of Microbiology. 1987;147(4):370-4.

21. Markou G, Georgakakis D. Cultivation of filamentous cyanobacteria (blue-green algae) in agro-industrial wastes and wastewaters: a review. Applied Energy. 2011;88(10):3389-401.

22. Berberoğlu H, Jay J, Pilon L. Effect of nutrient media on photobiological hydrogen production by Anabaena variabilis ATCC 29413. International Journal of Hydrogen Energy. 2008;33(4):1172-84.

23. Salleh SF, Kamaruddin A, Uzir MH, Mohamed AR. Effects of cell density, carbon dioxide and molybdenum concentration on biohydrogen production by Anabaena variabilis ATCC 29413. Energy Conversion and Management. 2014;87:599-605.

24. Weyman PD, Pratte B, Thiel T. Hydrogen production in nitrogenase mutants in Anabaena variabilis. FEMS microbiology letters. 2010;304(1):55-61.

25. Tsygankov A, Serebryakova L, Sveshnikov D, Rao K, Gogotov I, Hall D. Hydrogen photoproduction by three different nitrogenases in whole cells of Anabaena variabilis and the dependence on pH. International journal of hydrogen energy. 1997;22(9):859-67.

26. Tiwari A, Pandey A. Cyanobacterial hydrogen production–a step towards clean environment. International journal of hydrogen energy. 2012;37(1):139-50.

27. Dutta D, De D, Chaudhuri S, Bhattacharya SK. Hydrogen production by cyanobacteria. Microbial Cell Factories. 2005;4(1):1.

---

## [Decision Letter · Decision Letter 1]

27 Nov 2019

PONE-D-19-22664R1

Reconstruction of a regulated two-cell metabolic model to study biohydrogen production in a diazotrophic cyanobacterium Anabaena variabilis ATCC 29413

PLOS ONE

Dear Dr Ehsan Motamedian,

Thank you for submitting your revised manuscript to PLOS ONE. It was examined by four experts in the field, three recommanded acceptance of your paper but one of the reviewers has still some suggestions of amendments. Therefore, we invite you to submit a revised version of the manuscript that addresses the various points raised by this reviewers or to answer to his comments in an argumented way.

We would appreciate receiving your revised manuscript by end of december. To enhance the reproducibility of your results, we recommend that if applicable you deposit your laboratory protocols in protocols.io, where a protocol can be assigned its own identifier (DOI) such that it can be cited independently in the future. For instructions see: http://journals.plos.org/plosone/s/submission-guidelines#loc-laboratory-protocols

We look forward to receiving your revised manuscript.

Kind regards,

Marie-Joelle Virolle, PhD

Academic Editor

PLOS ONE

Reviewers' comments:

Reviewer's Responses to Questions

**Comments to the Author**

1. If the authors have adequately addressed your comments raised in a previous round of review and you feel that this manuscript is now acceptable for publication, you may indicate that here to bypass the “Comments to the Author” section, enter your conflict of interest statement in the “Confidential to Editor” section, and submit your "Accept" recommendation.

Reviewer #1: All comments have been addressed

Reviewer #2: All comments have been addressed

Reviewer #3: (No Response)

Reviewer #4: (No Response)

2. Is the manuscript technically sound, and do the data support the conclusions?

Reviewer #1: Yes

Reviewer #2: Yes

Reviewer #3: (No Response)

Reviewer #4: Yes

3. Has the statistical analysis been performed appropriately and rigorously? 

Reviewer #1: N/A

Reviewer #2: N/A

Reviewer #3: (No Response)

Reviewer #4: Yes

4. Have the authors made all data underlying the findings in their manuscript fully available?

Reviewer #1: No

Reviewer #2: Yes

Reviewer #3: (No Response)

Reviewer #4: Yes

5. Is the manuscript presented in an intelligible fashion and written in standard English?

Reviewer #1: Yes

Reviewer #2: Yes

Reviewer #3: (No Response)

Reviewer #4: Yes

6. Review Comments to the Author

Reviewer #1: Thanks for addressing all comments and suggestions, please make sure that you share the model as MAT, JSON, SBML, or XML.

Reviewer #2: Satisfied with replies and changes in the manuscript

Reviewer #3: (No Response)

Reviewer #4: Reconstruction of a two-cell metabolic model to study biohydrogen production in a diazotrophic cyanobacterium Anabaena variabilis ATCC 29413

Overview:---------------

In the manuscript titled "Reconstruction of a two-cell metabolic model to study biohydrogen production in a diazotrophic cyanobacterium Anabaena variabilis ATCC 29413", the authors prepared a describe genome-scale network reconstruction (GENRE) of Anabaena variabilis ATCC29413 and applied it to the increase production of hydrogen. In general, authors have included many changes while some critical results I think still need to included in the manuscript. This includes comparison of metabolic networks of their model and the previously published photosynthetic bacteria (iJN678 and iSynCJ816: both available through BiGG)

More importantly, the authors need to provide additional simulations and benchmark their models with existing models of other cyanobacteria or other two-cell models. The only benchmarks, Figure 1 and 2, are still quite opaque and not very informative of the model as a parameter for theirsimulations. I have suggested possible changes that make the figure clear.

Overall, keeping the above points in mind, I think the authors need to undergo major revisions of their manuscript. Please find my detailed comments below.

Comments:---------

Major:

In the revision, Table 1 compares the size of the model in this study to previously published models. It is possible I didn't clarify what I was looking for. I think the best way for the reader to understand their model is for the authors to show differences in the models. This includes showing genomic metabolic differences between different organisms. This goes beyond comparing the size of the models which is not sufficient is not an indicator of differences. Which pathways are complete/incomplete in Anabaena but complete in other organisms/models. I feel these need to be discussed. One way authors can accomplish this (only a suggestion, authors can improvise) is by drawing a schematic of specific pathways and differently coloring the reactions which are differentially present in their models versus other models. This is an easy way to show that their model does indeed capture biological differences between organisms characterized in their respective genomes. Further, given that their ids are in BiGG database, this analysis/comparison can be easily performed.

305-318: There are two points here:

a. The authors are talking about overprediction in high irradiance condition. Typically, models of photosynthetic organisms should behave as such when used with FBA: given the carbon uptake is constant, increase in light flux will increase growth upto a certain point, after that no amount of increase in light flux will cause any change in growth. Further, probably all of the existing models of photosynthetic organisms are likely to show this overprediction in high irradiance condition. After simulating the Two-cell model, I found that the model presented here captures both these aspects. The writeup in the manuscript represents the model in poor light. I would suggest that author find a different way of plotting Figure 2. One way to do this would be to adding two more subfigures in Figure 2: (1) Plot the light flux (x-axis) vs growth (y-axis) and find the ratio of uptake of light to HCO3; (2) Show what is this ratio in experimental conditions tested. Plotting this way does two things: (1) describes behavior of the model and (2) validates that behavior. Other models such as iJN678 and iSynCJ816 also do that. Overprediction in high irradiance condition means that organism reaches the plateau earlier than the model. This is expected as there likely gaps in the model; and hence, do not capture the entire range of bottlenecks. Discuss this point and show.

b. In essence, authors are generating context-specific model for two-cell metabolic network. The authors are capturing only the photosystem II gene (PsbJ). However, with changes in light, many other intracellular metabolic genes are likely to get differentially expressed; and thus, are likely to come up as hits for whats happening in the model. There are many transcriptomics studies in photosynthetic bacteria suggesting this. I would suggest that author explain why there's only one hit. Also I think it is important to have a figure for how the expression of various genes from photosystem II change across different light conditions; thus, reinforcing their argument.

I understand authors used TRFBA because they developed it. But I think authors should discuss if a different context-specific model extraction algorithm could reproduce the same results.

323-327: Please explain why the authors see this difference. What is the difference in the metabolic network of Malatinzsky et al and their model that yields transport of sucrose.

330: What are "both" substrates: Glutamine-glutamate or sucrose-(?)

331: Please explain this electron limitation and discuss the pathways the model uses to disspate excess electron at high irradiance.

352: Of course, disabling PSII for a photosynthetic organism will not lead to generation of oxygen. This is not surprising and has been known for a long time now. The authors need to discuss differences in metabolic network of vegetative state versus heterocyst state. I recommend that authors provide this comparison for the reader.

370-373: The mechanism described appears to be similar to how photosynthetic organisms deal with light-dark (diel) conditions. Is it the same? Are there any differences? Please discuss. What if my map A. variabilis genes to Synechocystis and use iJN678 to simulate A. variabilis behavior in combination with TRFBA. Will I get the same result. Again for this reason, the authors need to compare the metabolic network of their model with other photosynthetic bacteria. Synechocystis model is most well studied, so I would use that.

7. PLOS authors have the option to publish the peer review history of their article (what does this mean?). If published, this will include your full peer review and any attached files.

Reviewer #1: No

Reviewer #2: No

Reviewer #3: No

Reviewer #4: No

---

## [Author Response · Author response to Decision Letter 1]

12 Dec 2019

Thank you for your valuable comments. We revised the paper according to your suggestions. Here are our answers to your specified questions.

6. Review Comments to the Author

Reviewer #1: Thanks for addressing all comments and suggestions, please make sure that you share the model as MAT, JSON, SBML, or XML.

The mat-files of the single-cell, two-cell, and regulated two-cell models for implementation in MATLAB are presented in supplementary files S3.

Reviewer #2: Satisfied with replies and changes in the manuscript

Reviewer #3: (No Response)

Reviewer #4: Reconstruction of a two-cell metabolic model to study biohydrogen production in a diazotrophic cyanobacterium Anabaena variabilis ATCC 29413

Overview:---------------

In the manuscript titled "Reconstruction of a two-cell metabolic model to study biohydrogen production in a diazotrophic cyanobacterium Anabaena variabilis ATCC 29413", the authors prepared a describe genome-scale network reconstruction (GENRE) of Anabaena variabilis ATCC29413 and applied it to the increase production of hydrogen. In general, authors have included many changes while some critical results I think still need to included in the manuscript. This includes comparison of metabolic networks of their model and the previously published photosynthetic bacteria (iJN678 and iSynCJ816: both available through BiGG)

More importantly, the authors need to provide additional simulations and benchmark their models with existing models of other cyanobacteria or other two-cell models. The only benchmarks, Figure 1 and 2, are still quite opaque and not very informative of the model as a parameter for theirsimulations. I have suggested possible changes that make the figure clear.

Overall, keeping the above points in mind, I think the authors need to undergo major revisions of their manuscript. Please find my detailed comments below.

Comments:---------

Major:

In the revision, Table 1 compares the size of the model in this study to previously published models. It is possible I didn't clarify what I was looking for. I think the best way for the reader to understand their model is for the authors to show differences in the models. This includes showing genomic metabolic differences between different organisms. This goes beyond comparing the size of the models which is not sufficient is not an indicator of differences. Which pathways are complete/incomplete in Anabaena but complete in other organisms/models. I feel these need to be discussed. One way authors can accomplish this (only a suggestion, authors can improvise) is by drawing a schematic of specific pathways and differently coloring the reactions which are differentially present in their models versus other models. This is an easy way to show that their model does indeed capture biological differences between organisms characterized in their respective genomes. Further, given that their ids are in BiGG database, this analysis/comparison can be easily performed.

There are many different reactions between the reconstructed metabolic models of cyanobacteria. For example, there are 340 reactions in our metabolic model that are not present in iJN678 and there are 220 reactions in iJN678 that are not present in our metabolic model. So, it is not possible to indicate the differences by drawing a schematic of specific pathways.

305-318: There are two points here:

a. The authors are talking about overprediction in high irradiance condition. Typically, models of photosynthetic organisms should behave as such when used with FBA: given the carbon uptake is constant, increase in light flux will increase growth upto a certain point, after that no amount of increase in light flux will cause any change in growth. Further, probably all of the existing models of photosynthetic organisms are likely to show this overprediction in high irradiance condition. After simulating the Two-cell model, I found that the model presented here captures both these aspects. The writeup in the manuscript represents the model in poor light. I would suggest that author find a different way of plotting Figure 2. One way to do this would be to adding two more subfigures in Figure 2: (1) Plot the light flux (x-axis) vs growth (y-axis) and find the ratio of uptake of light to HCO3; (2) Show what is this ratio in experimental conditions tested. Plotting this way does two things: (1) describes behavior of the model and (2) validates that behavior. Other models such as iJN678 and iSynCJ816 also do that. Overprediction in high irradiance condition means that organism reaches the plateau earlier than the model. This is expected as there likely gaps in the model; and hence, do not capture the entire range of bottlenecks. Discuss this point and show.

It should be mentioned that the bicarbonate uptake rate is not constant according to the experimental data [1] and it increases in high irradiances. Based on your comment we plotted the light flux (x-axis) vs growth (y-axis) at constant bicarbonate uptake rate of 1 mmol/gDCW/h (Figure R1). Then, we found that the minimum photon uptake rate for optimal growth is about 9 mmol/gDCW/h. However, the ratio of experimental bicarbonate to photon uptake rates (Figure R2) demonstrates that the minimum ratio is 23.3 in the experimental condition. So, the photon uptake rate is always more than the required rate predicted by FBA. So, unfortunately, this data does not answer to overprediction while the calculated shadow prices predicted the reason for overprediction.

If there were gaps in the metabolic network, the model should underpredict the growth. However, the model overpredicted and hence, there were intracellular constraints. We used the regulated two-cell model and shadow price to predict the bottleneck gene(s) under high irradiances.

Figure R1. Growth rate versus photon uptake rates predicted by the two-cell model (bicarbonate uptake rate is limited to 1 mmol/gDCW/h).

Figure R2. Photon uptake rate per bicarbonate uptake rate for 18 experiments carried out by [1].

b. In essence, authors are generating context-specific model for two-cell metabolic network. The authors are capturing only the photosystem II gene (PsbJ). However, with changes in light, many other intracellular metabolic genes are likely to get differentially expressed; and thus, are likely to come up as hits for whats happening in the model. There are many transcriptomics studies in photosynthetic bacteria suggesting this. I would suggest that author explain why there's only one hit. Also I think it is important to have a figure for how the expression of various genes from photosystem II change across different light conditions; thus, reinforcing their argument.

We used the easiest way and calculated shadow prices for all genes. Shadow price indicates the effect of one unit increase in gene expression on the predicted growth rate. Shadow price for other genes was zero and hence, the regulated two-cell model predicted that the measured expression level for other genes was adequate for growth. According to your comment, we plotted the growth rate versus expression level for some genes of photosystem II. It can be seen that only the expression level of PsbJ was not sufficient for optimal growth.

Figure 3. Predicted growth rate versus expression level for some genes of photosystem II using the regulated two-cell model. The measured expression level [2] for each gene is determined by a circle in all of the figures.

The explanations were added to the manuscript.

I understand authors used TRFBA because they developed it. But I think authors should discuss if a different context-specific model extraction algorithm could reproduce the same results.

TRFBA adds a constraint (Eq. 4) to the model for each gene with the measured expression level. So, we could use shadow price to indicate the effect of the expression level of each gene on growth and we could easily find the bottleneck gene. The predicted photoinhibition was in accordance with previous reports [3-6].

Our main aim in this research was the reconstruction and evaluation of a genome-scale metabolic model for A. variabilis. Evaluation of the integration algorithms is not our aim in this research. Some benchmarks are developed for evaluation of the integration algorithms (e. g. [7-9]). In our previous works [8, 10], we indicated that TRFBA is a successful algorithm for prediction of growth.

323-327: Please explain why the authors see this difference. What is the difference in the metabolic network of Malatinzsky et al and their model that yields transport of sucrose.

Interestingly, glutamate and sucrose intercellular exchange reactions are coupled in the two-cell model of Malatinzsky et al. [11] and the sucrose rate is maximized when glutamate is exchanged at the maximum rate. Furthermore, Figure S4 shows that maximum exchange rates for this two-cell model are always higher than those for our two-cell models and the rates unreasonably increase with an enhancement of irradiance. The maximal sucrose uptake rate of 46.8 mmol/gDCW/h is calculated at an irradiance of 16100 lux and bicarbonate uptake rate of 2.14 mmol/gDCW/h while this rate for our two-cell and regulated two-cell models is 1.29 and 1.12 mmol/gDCW/h, respectively. This indicates that there is an internal cycle of carbon in the two-cell model of Malatinzsky et al. [11]. This cycle circulates carbon sources between heterocyst and vegetative cells and the required energy for this circulation is provided by uptake of photon. Comparison between flux distributions for maximum and minimum intercellular exchange rates indicates that many reactions are involved in generating this internal cycle. Hence, Malatinzsky et al. [11] fixed the glutamine-glutamate exchange ratio to one that results in the reduction of the effect of this cycle.

The explanations were added to the manuscript.

330: What are "both" substrates: Glutamine-glutamate or sucrose-(?)

Glutamate and sucrose are correct. The manuscript was clarified.

331: Please explain this electron limitation and discuss the pathways the model uses to disspate excess electron at high irradiance.

We mean the limitation of PSII that previously explained. The manuscript was clarified.

As mentioned in the material and methods section, the maximum uptake rate of photon was set to the experimental values [1] because more irradiance does not necessarily increase the photon uptake rate. So, the model consumed photon with the required rate.

352: Of course, disabling PSII for a photosynthetic organism will not lead to generation of oxygen. This is not surprising and has been known for a long time now. The authors need to discuss differences in metabolic network of vegetative state versus heterocyst state. I recommend that authors provide this comparison for the reader.

This sentence is used for further emphasis. In the next sentences, we also explained more about the reason for the difference in the metabolic network of vegetative cells and heterocysts for oxygen. We also drew a schematic of the metabolic pathways and reactions in Figure 4 that shows the differences in the metabolic networks of heterocysts and vegetative cells.

370-373: The mechanism described appears to be similar to how photosynthetic organisms deal with light-dark (diel) conditions. Is it the same? Are there any differences? Please discuss. What if my map A. variabilis genes to Synechocystis and use iJN678 to simulate A. variabilis behavior in combination with TRFBA. Will I get the same result. Again for this reason, the authors need to compare the metabolic network of their model with other photosynthetic bacteria. Synechocystis model is most well studied, so I would use that.

Yes, it is the same and similar results for Synechocystis using its metabolic model (iJN678) are obtained without using TRFBA or mapping genes of A. variabilis to Synechocystis. This result is rational for every metabolic model of cyanobacteria because fructose provides both energy and carbon sources for the cell. However, in photoautotrophic conditions, bicarbonate is used as a carbon source and photon uptake that is necessary for energy generation produces oxygen.

The explanations were added to the manuscript.

Reference

1. Berberoğlu H, Barra N, Pilon L, Jay J. Growth, CO₂ consumption and H₂ production of Anabaena variabilis ATCC 29413-U under different irradiances and CO₂ concentrations. Journal of applied microbiology. 2008.

2. Park J-J, Lechno-Yossef S, Wolk CP, Vieille C. Cell-specific gene expression in Anabaena variabilis grown phototrophically, mixotrophically, and heterotrophically. BMC genomics. 2013;14(1):759.

3. Yoon JH, Sim SJ, Kim M-S, Park TH. High cell density culture of Anabaena variabilis using repeated injections of carbon dioxide for the production of hydrogen. International journal of hydrogen energy. 2002;27(11-12):1265-70.

4. Yoon JH, Shin J-H, Park TH. Characterization of factors influencing the growth of Anabaena variabilis in a bubble column reactor. Bioresource technology. 2008;99(5):1204-10.

5. Agel G, Nultsch W, Rhiel E. Photoinhibition and its wavelength dependence in the cyanobacterium Anabaena variabilis. Archives of Microbiology. 1987;147(4):370-4.

6. Markou G, Georgakakis D. Cultivation of filamentous cyanobacteria (blue-green algae) in agro-industrial wastes and wastewaters: a review. Applied Energy. 2011;88(10):3389-401.

7. Machado D, Herrgård M. Systematic evaluation of methods for integration of transcriptomic data into constraint-based models of metabolism. PLoS computational biology. 2014;10(4):e1003580.

8. Jamialahmadi O, Hashemi-Najafabadi S, Motamedian E, Romeo S, Bagheri F. A benchmark-driven approach to reconstruct metabolic networks for studying cancer metabolism. PLoS computational biology. 2019;15(4):e1006936.

9. Pacheco MP, Pfau T, Sauter T. Benchmarking procedures for high-throughput context specific reconstruction algorithms. Frontiers in physiology. 2016;6:410.

10. Motamedian E, Mohammadi M, Shojaosadati SA, Heydari M. TRFBA: an algorithm to integrate genome-scale metabolic and transcriptional regulatory networks with incorporation of expression data. Bioinformatics. 2016;33(7):1057-63.

11. Malatinszky D, Steuer R, Jones PR. A comprehensively curated genome-scale two-cell model for the heterocystous cyanobacterium Anabaena sp. PCC 7120. Plant physiology. 2017;173(1):509-23.

---

## [Decision Letter · Decision Letter 2]

23 Dec 2019

PONE-D-19-22664R2

Reconstruction of a regulated two-cell metabolic model to study biohydrogen production in a diazotrophic cyanobacterium Anabaena variabilis ATCC 29413

PLOS ONE

Dear Dr. Motamedian,

Thank you for submitting your revised manuscript to PLOS ONE. After careful consideration by two of the three previous reviewers, we feel that it has merit but does not fully meet PLOS ONE’s publication criteria as it currently stands. Therefore, we invite you to submit a revised version of the manuscript that addresses the rather minor points raised during the review process.

We would appreciate receiving your revised manuscript by end of january. To enhance the reproducibility of your results, we recommend that if applicable you deposit your laboratory protocols in protocols.io, where a protocol can be assigned its own identifier (DOI) such that it can be cited independently in the future. For instructions see: http://journals.plos.org/plosone/s/submission-guidelines#loc-laboratory-protocols

We look forward to receiving your revised manuscript.

Kind regards,

Marie-Joelle Virolle, PhD

Academic Editor

PLOS ONE

Reviewers' comments:

Reviewer's Responses to Questions

**Comments to the Author**

1. If the authors have adequately addressed your comments raised in a previous round of review and you feel that this manuscript is now acceptable for publication, you may indicate that here to bypass the “Comments to the Author” section, enter your conflict of interest statement in the “Confidential to Editor” section, and submit your "Accept" recommendation.

Reviewer #2: All comments have been addressed

Reviewer #4: (No Response)

2. Is the manuscript technically sound, and do the data support the conclusions?

Reviewer #2: Yes

Reviewer #4: Yes

3. Has the statistical analysis been performed appropriately and rigorously? 

Reviewer #2: N/A

Reviewer #4: Yes

4. Have the authors made all data underlying the findings in their manuscript fully available?

Reviewer #2: Yes

Reviewer #4: Yes

5. Is the manuscript presented in an intelligible fashion and written in standard English?

Reviewer #2: Yes

Reviewer #4: Yes

6. Review Comments to the Author

Reviewer #2: No additional comments

Reviewer #4: Reconstruction of a two-cell metabolic model to study biohydrogen production in a diazotrophic cyanobacterium Anabaena variabilis ATCC 29413

Overview:---------------

The authors have for the most part addressed all of my previous concerns. But I think the one analysis that I reiterate here are important addition to this paper and contains important information about the meabolic model reconstructed here. Keeping this in mind, I recommend minor revisions.

Comments:---------

Major:

Clearly, the authors agree that there are difference between their model and iJN678. As I had stated in the previous round that pathway specific differences need to be discssed between their model and existing model(s). Since, this is the first time this model is being published, it comes on the authors to elaborate on this. The authors only need pathway annotations from the models to prepare a bar graph where each bar represents one pathway. This is clearly possible and relatively quickly.

7. PLOS authors have the option to publish the peer review history of their article (what does this mean?). If published, this will include your full peer review and any attached files.

Reviewer #2: No

Reviewer #4: No

---

## [Author Response · Author response to Decision Letter 2]

27 Dec 2019

Thank you for your valuable comments. We revised the paper according to your suggestions. Here are our answers to your specified questions.

Review Comments to the Author

Reviewer #2: No additional comments

Reviewer #4: Reconstruction of a two-cell metabolic model to study biohydrogen production in a diazotrophic cyanobacterium Anabaena variabilis ATCC 29413

Overview:---------------

The authors have for the most part addressed all of my previous concerns. But I think the one analysis that I reiterate here are important addition to this paper and contains important information about the meabolic model reconstructed here. Keeping this in mind, I recommend minor revisions.

Comments:---------

Major:

Clearly, the authors agree that there are difference between their model and iJN678. As I had stated in the previous round that pathway specific differences need to be discssed between their model and existing model(s). Since, this is the first time this model is being published, it comes on the authors to elaborate on this. The authors only need pathway annotations from the models to prepare a bar graph where each bar represents one pathway. This is clearly possible and relatively quickly.

According to your comment, we carefully matched the reactions of the two models and eliminated the same reactions under different names. Two bar charts (Figures S1 (a) and (b), Supplementary file S4) were plotted to indicate the distribution of different reactions in the metabolic pathways. Furthermore, more explanations were added to the manuscript.

---

## [Decision Letter · Decision Letter 3]

6 Jan 2020

Reconstruction of a regulated two-cell metabolic model to study biohydrogen production in a diazotrophic cyanobacterium Anabaena variabilis ATCC 29413

PONE-D-19-22664R3

Dear Dr. ,Ehsan Motamedian

We are pleased to inform you that your manuscript has been judged scientifically suitable for publication and will be formally accepted for publication once it complies with all outstanding technical requirements.

With kind regards,

Marie-Joelle Virolle, PhD

Academic Editor

PLOS ONE

Additional Editor Comments (optional):

Reviewers' comments:

Reviewer's Responses to Questions

**Comments to the Author**

1. If the authors have adequately addressed your comments raised in a previous round of review and you feel that this manuscript is now acceptable for publication, you may indicate that here to bypass the “Comments to the Author” section, enter your conflict of interest statement in the “Confidential to Editor” section, and submit your "Accept" recommendation.

Reviewer #4: All comments have been addressed

2. Is the manuscript technically sound, and do the data support the conclusions?

Reviewer #4: Yes

3. Has the statistical analysis been performed appropriately and rigorously? 

Reviewer #4: Yes

4. Have the authors made all data underlying the findings in their manuscript fully available?

Reviewer #4: Yes

5. Is the manuscript presented in an intelligible fashion and written in standard English?

Reviewer #4: Yes

6. Review Comments to the Author

Reviewer #4: (No Response)

7. PLOS authors have the option to publish the peer review history of their article (what does this mean?). If published, this will include your full peer review and any attached files.

Reviewer #4: No

---

## [Editor Report · Acceptance letter]

16 Jan 2020

PONE-D-19-22664R3 

Reconstruction of a regulated two-cell metabolic model to study biohydrogen production in a diazotrophic cyanobacterium *Anabaena variabilis* ATCC 29413 

Dear Dr. Motamedian:

I am pleased to inform you that your manuscript has been deemed suitable for publication in PLOS ONE. Congratulations! Your manuscript is now with our production department. 

With kind regards,

on behalf of

Dr. Marie-Joelle Virolle 

Academic Editor

PLOS ONE